# Structure, Fractality, Mechanics and Durability of Calcium Silicate Hydrates

**Shengwen Tang** [1,2,3,4,5,*], **Yang Wang** [1], **Zhicheng Geng** [1], **Xiaofei Xu** [1], **Wenzhi Yu** [6], **Hubao A** [1] **and Jingtao Chen** [1]

1   State Key Laboratory of Water Resources and Hydropower Engineering Science, Wuhan University, Wuhan 430072, China; 2020202060084@whu.edu.cn (Y.W.); 2017301580356@whu.edu.cn (Z.G.); xuxiaofei@whu.edu.cn (X.X.); a_hubao@whu.edu.cn (H.A.); jingtao_chen3@163.com (J.C.)
2   State Key Laboratory of Building Safety and Built Environment, Beijing 100013, China
3   National Engineering Research Center of Building Technology, Beijing 100053, China
4   State Key Laboratory of Green Building Materials, China Building Materials Academy, Beijing 100024, China
5   Suzhou Institute of Wuhan University, Suzhou 215123, China
6   State Key Laboratory for Health and Safety of Bridge Structures, China Railway Bridge Science Research Institute CO., Ltd., Wuhan 430034, China; ywz0904@126.com
*   Correspondence: tangsw@whu.edu.cn

**Abstract:** Cement-based materials are widely utilized in infrastructure. The main product of hydrated products of cement-based materials is calcium silicate hydrate (C-S-H) gels that are considered as the binding phase of cement paste. C-S-H gels in Portland cement paste account for 60–70% of hydrated products by volume, which has profound influence on the mechanical properties and durability of cement-based materials. The preparation method of C-S-H gels has been well documented, but the quality of the prepared C-S-H affects experimental results; therefore, this review studies the preparation method of C-S-H under different conditions and materials. The progress related to C-S-H microstructure is explored from the theoretical and computational point of view. The fractality of C-S-H is discussed. An evaluation of the mechanical properties of C-S-H has also been included in this review. Finally, there is a discussion of the durability of C-S-H, with special reference to the carbonization and chloride/sulfate attacks.

**Keywords:** calcium silicate hydrate; fractal; calcium to silicate ratio; mechanical properties; carbonization; chloride/sulfate attack



## 1. Introduction

The mechanical properties and durability of cement-based materials are of great significance. Mechanical properties affect the safety of building structures, and durability is associated with the service life of buildings. In terms of mechanical properties, the compressive strength of cement-based materials is quite high. However, their tensile strength is quite low, usually only 10−15% of their compressive strength [1,2], which leads to the fact that cement-based materials need to work in conjunction with steel bars with excellent tensile properties when they are utilized in the engineering construction. The durability of cement-based materials refers to its ability to resist the effects of environmental media and maintain its good performance and appearance integrity for a long time, thereby maintaining the safe and normal use of materials. The surrounding environment of the concrete structure will have some adverse effects on its durability. For instance, carbon dioxide in the air will reduce the alkalinity of cement-based materials, which will weaken the antirust protection of concrete to steel reinforcement and easily cause corrosion of steel reinforcement. High durability of cement-based materials will be required in some special environments. For instance, in cold areas, cement-based materials will undergo freezing–thawing cycles, resulting in a significant reduction in strength, generally accompanied by surface looseness, creep, strip and even the leakage of reinforcement [3].Additionally, a variety of ions in the marine and saline-alkali environment will have an erosion effect on

cement-based materials: sulfate attack will cause cement-based materials expansion and decomposition; chloride ions attack will easily lead to steel corrosion [4].

Calcium silicate hydrate (C-S-H) is the main hydration product of Portland cement [5–7], occupying 60–70% of the total volume of products [8–10], and is often considered as the key component of cement-based materials affecting their mechanical properties and durability [11]. Furthermore, the structure of C-S-H can directly affect the strength, shrinkage and creep of the cement paste [12–16].Therefore, it has a quite important guiding significance on understanding the micro-nano structure of C-S-H, as well as improving the mechanical properties of cement-based materials. Cement-based materials are the most used man-made materials in the world, and C-S-H is the key factor that determines their mechanical strength and durability. Therefore, the accurate understanding of the micromechanical properties of C-S-H is the key to improving the mechanical properties of cement-based materials.

It has been widely suggested that cement-based material is a kind of fractal [17–21], of which the pore structure exhibits local and global self-similarity [22]. Surface and volume (mass) fractal dimensions are usually used to describe the fractal texture of pore structure in cement-based materials. Surface fractality means that the porous material has an irregular self-similar pore surface, while volume fractality describes the spatial scaling behavior (volume). The value of volume fractal dimension is generally higher than 3, while the one of surface fractal dimension is between 2 and 3. The hardened cement paste has quite complex and disordered pore structure, which is difficult to be described by classical Euclidean parameters [23–25].

Carbonization is one of factors affecting the durability of cement-based materials. It is said that the main carbonization focuses on the alkaline substance in cement hydration product. The carbonization of C-S-H has not received enough attention; it is said that the carbonization investigation of C-S-H is of great significance to improve the durability of cement-based materials. Additionally, in recent years, the accelerated carbonization curing method has begun to be applied to the production of precast concrete. This process can absorb a large amount of carbon dioxide and, thus, has important environmental protection significance. The chloride/sulfate attack is another factor affecting the durability of cement-based materials. In this work, the comparison of preparation methods of C-S-H is introduced. Structure of C-S-H is analyzed comprehensively. The fractality of C-S-H gels is discussed. Mechanical properties of C-S-H will be summarized from the view point of experiments and simulations. Finally, the durability of C-S-H associated with carbonization and under chloride/sulfate attacks is reviewed.

## 2. Materials and Methods

### 2.1. Hydration of Tricalcium Silicate and Dicalcium Silicate

The hydration reaction of Portland cement is a heterogeneous multiphase chemical reaction process. The hydration products mainly consist of C-S-H, calcium hydroxide (CH), ettringite (AFt), aluminate-ferrite-monosubstituted phases (AFm) and unhydrated cement particles. These hydration products are interlinked, whose types and quantities change over time. To eliminate the interference of other phases in cement hydration products and obtain C-S-H samples as purely as possible, the utilization of $C_3S$ seems a good choice to produce C-S-H. Such C-S-H specimens obtained in this way perhaps fully demonstrate the in situ state of C-S-H in cement paste.

The only hydration products of tricalcium silicate ($C_3S$) or dicalcium silicate ($C_2S$) are C-S-H and CH, which simplifies the analysis of C-S-H. From results from nitrogen and water vapor adsorption tests, the C-S-H gel in the hardened $C_3S$ paste has two kinds of pores, namely, wide inter-gel pores and small inner pores [26]. Lawrence et al. from electron microscopy examinations and found that C-S-H had two different forms: one appeared as needle-like, and the other could not be distinguished by the microscope [27]. Seishi et al. combined scanning electron microscope (SEM) and electron probe micro-analyzer to further analyze the difference between the internal and external textures of C-S-H gels; according to the difference of Ca/Si ratio, the morphology of C-S-H was divided into two

types: when the ratio of Ca/Si was larger than 3, C-S-H had a smooth surface, while it showed needle-shaped, and vice versa [28]. Ciach et al. and Jennings et al. used SEM to analyze the morphology of the generated C-S-H with the hydration time; the plates appeared to have an underlying mesh structure of interlocking fibers, the space between these had been filled with a lime-rich C-S-H or with CH [29,30]. Stucke found that in the hydrating $C_3S$ paste the edge of the hydration product consisted of two layers of material, the outer part of the needle-like formed with a Ca/Si ratio of 1.6 ± 0.1 and the inner C-S-H with a Ca/Si ratio of 1.90 ± 0.05 [31]. Alizadeh et al. added C-S-H seeds during the hydration process of $C_3S$ paste; the results showed that C-S-H seeds could significantly increase the dissolution rate and degree of dissolution of $C_3S$ particles and promoted the polymerization of C-S-H and silicate [32]. The properties of the resulting C-S-H gels depended on the ratio of CaO to $SiO_2$ in the raw material. Sato et al. studied the effect of nano-$CaCO_3$ on $C_3S$ hydration [33]. C-S-H growth was observed around $CaCO_3$ particles. The seeding effect produced by adding nano-$CaCO_3$ promoted $C_3S$ hydration and greatly shortened the induction period of $C_3S$ hydration. Begarin et al. synthesized alite with an aluminum content of 0.1 wt% to study the role of aluminum in the hydration reaction of $C_3S$. C-S-H containing aluminum ions was formed at early age [34]. This hydrate caused the growth of C-S-H to be delayed before hydration was accelerated. Fernandez et al. studied the influence of magnesium on the hydration of $C_3S$ and found that C-S-H gels synthesized by $C_3S$ in the presence of MgO had different Ca/Si ratios [35]. Rodriguez et al. studied the influence of the concentration of CaO in the solution on the morphology of C-S-H during the $C_3S$ reaction. The study found that when the concentration of CaO was lower than 22 mmol/L, the synthesized sample was in the shape of a foil [36]. As the concentration of CaO increased, the morphology of the C-S-H gel was fibrous transition; nuclear magnetic resonance spectroscopy (NMR) detection revealed an increase in $Q^1$ in the sample. Garralult et al. quantified the growth of C-S-H parallel to and perpendicular to the alite surface through atomic force microscope test and numerical simulation and found that the growth rate of C-S-H gels only depended on the concentration of CaO in the solution [37]. José et al. comparatively studied the hydration behavior of $C_3S$ in water and NaOH solutions [38]. The over-alkaline medium had little significant effect on the mechanical development of $C_3S$ hydration products, but the alkaline environment accelerated the hydration of $C_2S$ and affected the setting and hardening time. The reactions produced C-N-S-H gel, in which some Na atoms replaced Ca in the structure skeleton.

Although the hydration of a single cement mineral can reduce the variety of cement hydration products to some extent, the hydration products of $C_3S$ or $C_2S$ will still include CH and unhydrated particles. The C-S-H obtained through hydration reaction is usually interwoven with other substances, the chemical composition of C-S-H is not fixed, and the proportions of calcium and silicon elements fluctuate at a certain range. At present, the accuracy of evaluation techniques may not be enough to select exactly C-S-H materials with high purity from cement hydration products under complex conditions. With the increasing attention paid to C-S-H, the research field of C-S-H has been also greatly expanded, and some chemical methods have been introduced into the preparation of C-S-H and gradually developed and matured. To avoid the interference of other substances, some other chemical synthesis methods have been introduced into the fabrication of pure C-S-H. At present, the most commonly utilized methods are sol-gel, chemical coprecipitation and hydrothermal synthesis methods.

### *2.2. Sol-Gel Method*

The sol-gel method hydrolyzes the metal-organic alcohol salt, the metal inorganic salt or their mixture to form the sol. The sol is gradually gelled through aging polymerization, and the gel is dried and calcined to obtain the inorganic material. The sol-gel method has been newly developed in recent years and can replace the high-temperature calcination method to prepare glass, ceramics and other oxide solid materials. The sol-gel method has the advantages of low temperature required for product preparation, easy control of the

chemical composition of products, uniform addition of trace components, good uniformity and high purity of synthetic products. Therefore, the sol-gel method has been widely utilized in the fabrication of ultra-fine powder materials and nano-oxide films.

C-S-H and polymer nanocomposites with certain content of polyacrylic acid were synthesized by continuous agitation under nitrogen conditions and continuous agitation of suspension at 60 °C for 7 days. Minet et al. synthesized a new organic–inorganic C-S-H hybrid material by the sol-gel method in an alkaline environment [39]. The results showed that the synthesized C-S-H intermediate layer contained organic parts, but the organic compounds did not destroy the inorganic skeleton structure of C-S-H. Franceschini et al. synthesized C-S-H compounds containing both trialkoxysilane and methyldialkoxysilane by the sol-gel method. Furthermore, results from $^{29}$Si CP magic angle spinning NMR technology (MAS-NMR) confirmed there were covalent bonds between the silicate chains of C-S-H and trialkoxysilane functional groups. This kind of organic polymer modified C-S-H materials could improve the mechanical properties and durability of the cement-based material. In the preparation of samples by the sol-gel method, dispersants are usually used to reduce the viscosity of the solution. Therefore, C-S-H prepared by this method has a high uniformity at the molecular level. However, C-S-H fabricated by this method is quite different from the case of cement hydration and takes a long time for the preparation, so the application of the sol-gel method is usually confined in the laboratory.

### 2.3. Chemical Coprecipitation Method

The chemical coprecipitation method refers to the process in which an insoluble substance formed in a chemical reaction causes another soluble substance to precipitate together, which can make multiple components precipitated at the same time, and the distribution of each component is relatively uniform and the fraction is relatively constant. Chemical coprecipitation method has the advantages of simple operation, low cost, short cycle, low temperature required and easy control, etc. It has been widely used to fabricate fine powder in laboratory or industrial scenarios. Suzuki et al. prepared C-S-H at room temperature using $CaCl_2$ and $H_4SiO_4$ solution as raw materials by the chemical coprecipitation method [40] and found that with the increase in the Ca/Si ratio, the morphology of the synthesized C-S-H specimens changed from microbeads to slightly spherical or blocky particles, which was composed of many small plate-like aggregates of microcrystals. Russias et al. stored the suspensions of CaO, $SiO_2$ and $AlNaO_2$ in $N_2$ at 20 °C and then stirred them continuously in a sealed polypropylene bottle for 3 weeks [41]. After washing and drying, the precipitation was determined to be the C-S-H material doped with Al (C-A-S-H). The chemical coprecipitation method is an effective way to prepare C-S-H with organic polymer materials doped. Pelisser et al. pre-dissolved polyvinyl alcohol (PVA) in $CO_2$-free deionized water [42], then mixed $CaNO_3$ solution with $NaNO_3$ solution gradually and prepared C-S-H polymer nanocomposites through chemical coprecipitation method. The results showed that organics embedded in the C-S-H interlayer, and the micromechanical properties of C-S-H changed significantly. Mojumdar et al. also fabricated C-S-H polymer nanocomposites (C-S-HPN) with PVA doped by chemical coprecipitation method [43]. X-ray diffraction (XRD) analysis showed that there were both PVA interlayers and C-S-H spallation in the intermediate tissues of C-S-HPN materials, and the SEM morphology of C-S-HPN materials with different PVA contents was significantly different.

### 2.4. Hydrothermal Synthesis Method

The hydrothermal synthesis method involves using any one of the many techniques to crystallize substances. The reaction temperature of the hydrothermal method is usually between 100 and 250 °C, and the steam pressure of water is between 0.3 and 4 MPa. Crystallized powder materials can be obtained directly without high-temperature calcinations, effectively avoiding the agglomeration of particles. The hydrothermal method can partially replace the high-temperature solid reaction, due to significantly reducing the temperature required for the reaction, good crystalline, high purity and good disparity for the fabricated

materials. Therefore, the hydrothermal method is commonly used for the fabrication of fine crystalline powder materials.

In recent years, autoclaved cured concrete has been promoted, which not only accelerates the growth rate of concrete strength, but also improves the ultimate strength of concrete. It is widely used in some projects that require high early strength of concrete and high turnover rate of formwork, especially in the production of precast concrete. The strength development of steam-cured concrete is closely related to the formation of C-S-H [44,45]. Inspired by this curing method, some researchers have prepared various forms of C-S-H powder materials by the hydrothermal synthesis method. Mörtel successfully synthesized C-S-H from $SiO_2$ and CH under the hydrothermal conditions of 180−200 °C and 16 bar saturated vapor pressure [46], respectively. Okada et al. prepared C-S-H with Ca/Si ratio of 0.3–2.0 in the stirred suspension at 120–180 °C with the mixture of CaO and $H_2SiO_3$ as raw materials [47]. Among them, when the reaction took a long time at 150−180 °C, amorphous C-S-H would further react to form crystalline products, especially when the Ca/Si ratio was 0.8, 11 Å tobermorite crystals would be formed. Števula et al. found that $\beta$-$C_2S$ or $\gamma$-$C_2S$ and silica suspension reacted under 190 °C hydrothermal condition for 8 h would produce C-S-H mesophase with interlamellar spacing of 16 Å [48]. If the reaction time was extended, C-S-H would turn into gyrolite. Siauciunas et al. also found that in the process of fabricating gyrolite through hydrothermal synthesis method [49], the intermediate compound C-S-H (I) would always be formed, especially when the ratio of Ca/Si was close to 0.8, the stable C-S-H with the interlamellar spacing of 11.3 Å would be formed.

The hydrothermal synthesis method is an effective way to prepare crystal materials, which is prominent in the preparation of ion-doped tobermorite materials. Diamond et al. prepared Al-doped tobermorite by the hydrothermal synthesis method [50], and X-ray fluorescence spectrum analysis was used to determine the position of Al in the tetrahedron. Surface area measurement showed that the inclusion of Al slightly reduced the particle size of tobermorite, and the substituted products showed strong characteristics of the high-temperature exothermic response. Nocuń-Wczelik et al. mixed CH, pure silica gel and water to fabricate C-S-H under the saturated steam pressure of 200 °C [51]. The influence of $CaCl_2$, $AlCl_3$, $CrCl_3$, $Na_2CrO_4$, NaOH and $Al(OH)_3$ on the formation of C-S-H was studied. The results indicated that Al had a stabilizing effect on tobermorite; $CrCl_3$ could promote the formation of disordered C-S-H with high water content, surface area and porosity; $CaCl_2$ and NaOH contributed to the formation of xonotlite. Mostafa et al. fabricated C-S-H with $Fe^{3+}$ and $Mg^{2+}$ substituents contained by the hydrothermal synthesis method at 175 °C [44]. It was found that $Mg^{2+}$ could improve the crystallinity of tobermorite and extend the time of the treatment of hydrothermal. The morphology of tobermorite changed from plate-like to layer-like. $Fe^{3+}$ increased the defects of tobermorite in a short time of the treatment of hydrothermal but increased its crystallinity during a long time of hydrothermal treatment. The crystal morphology of tobermorite with Fe doping changed from network-like to fiber-like. Miyake et al. fabricated tobermorite specimens with Al + Na and Al + K double-doped, respectively, through hydrothermal synthesis method. They subsequently studied the ion exchange properties of tobermorite for $K^+$ and $Cs^+$ from the perspectives of dynamic, equilibrium and thermodynamics. It was proved that the activation energy of Na to K was less than that of Na to Cs. The fabricated tobermorite was selective to $Cs^+$ ions, so it can be used for the exchange separation of cation [52].

In recent years, chemical additives have played a crucial role in the preparation of materials. Some scholars have also introduced chemical additives into the fabrication of C-S-H gels through the hydrothermal synthesis method. Hartmann et al. used $CaCO_3$ calcined at 1000 °C for 3 h as calcium sources [53,54] and fine $SiO_2$ powder or coarse quartz sand as a silicon source and produced C-S-H through a hydrothermal reaction at 200 °C. The effects of sucrose and calcium formate as additives on the crystallization of C-S-H were studied, respectively. The results showed that sucrose changed the reaction mechanism of the C-S-H system as an additive, and the hydrothermal process started from the rapid

reaction of sucrose and CaO, and the stable hydrothermal decomposition products formed had a strong inhibitory effect on the formation rate of C-S-H gels. Mostafa et al. used CH, finely dispersed aerosol (amorphous $SiO_2$) and ethylene diamine tetra-acetic acid as the calcium source under the pressure of 2 MPa, temperature of 200 °C, silicon source and a complexing agent [55]. Tobermorite fiber was synthesized successfully through hydrothermal reaction for 3−10 h. The length of the fiber was 40–100 μm, the aspect ratio was from 20 to 80 and the shape of the fiber could keep stable under 1150 °C. Some scholars have improved the original hydrothermal synthesis method for fabricating C-S-H. The comparison of preparation methods of C-S-H is demonstrated in Table 1.

**Table 1.** Comparison of preparation methods of C-S-H.

| Method | Principle | Advantages | Disadvantages |
| --- | --- | --- | --- |
| Hydration of $C_3S$ and $C_2S$ [26–38] | Use a single $C_3S$ hydration method to produce C-S-H | Reduces the types and quantities of hydration products | $Ca(OH)_2$ exists; unhydrated particles found |
| Sol-gel method [39] | Metal salt decomposed, sol formed, gelled, inorganic material obtained | Low temperature, easy control, uniformly trace components, synthesis product uniformity and high purity | Long time to prepare, limited to the laboratory |
| Chemical coprecipitation method [40–43] | Insoluble substance causes another soluble substance to precipitate together | Simple synthesis operation, low cost, short synthesis cycle, low synthesis temperature, conditions easy to control | Precipitant addition causes high local concentration, agglomeration, uneven composition |
| Hydrothermal synthesis method [44–55] | High-temperature pressure, material undergoes hydrolysis reaction, crystals formed when supersaturated | Replace high-temperature solid-phase reaction; reduce reaction temperature; product good crystallinity, high purity, good dispersion | Low yield, long reaction time |

## 3. C-S-H Structure

### 3.1. Study on the Structure of C-S-H

Understanding the molecular structure of materials is the basis for the study and improvement of the properties of materials. Therefore, the study of the molecular structure of C-S-H has a profound guiding significance for improving the mechanical properties and durability of cement-based materials. C-S-H presents obvious amorphous characteristics at the macro level because no clear crystal structure can be observed, making it difficult to obtain the specific molecular structure information of C-S-H. However, some early studies suggested that when the dimension was reduced to the atomic or nanoscale, the crystallinity of C-S-H was greatly improved due to the presence of nanoscale crystallinity and a short ordered range [56,57]. The current analysis of the molecular structure C-S-H mainly benefits from the rapid development of advanced testing techniques. The obtainment of the structural characteristics of C-S-H through various experimental detection techniques is also the basis of its molecular structure modeling. The rapid development of electron microscopy in the 1950s results in a large number of observations of the morphology of C-S-H. It is considered that there are four types of gel morphology of C-S-H observed in Portland cement paste by SEM [58]. Type I are fibrous particles with a length ranging from 0.5 to 2 μm, which is abundant in the early cement paste. Type II are network-like particles, which are not found in the hydration products of $C_3S$ or $C_2S$. Type III are large particles with a diameter of about 300 nm. The C-S-H of the above three morphologies are all outer products, and the internal products are defined as the C-S-H of Type IV. He et al. fabricated C-S-H with Ca/Si ratio of 1.0–1.7 and studied the influence of the ratio of Ca/Si on the morphology and structure of C-S-H [59]. SEM detection results showed that with the increase in the Ca/Si ratio, the polymerization degree of the silicon-oxygen tetrahedron in C-S-H decreased, and the morphology of C-S-H changed from plate to fiber with net shape. The determination of chemical composition is the key to the establishment of a molecular structure model. Additionally, the important stoichiometric parameter to

define the C-S-H phase is the molar ratio of CaO and $SiO_2$ in its structure (i.e., the ratio of calcium to silicate).

In the early 1970s, Diamond utilized SEM and energy dispersive X-ray spectrometer (EDS) together. The research results believed that the Ca/Si ratio of the local C-S-H region of hardened cement paste was within the range of 2 to 3 [60]. Besides, other studies have shown that when the ratio of water to cement (W/C) ratio is in the range of 0.4 to 0.7 and the ambient temperature is 20–25 °C, the Ca/Si ratio of the internal hydration product C-S-H in the hardened cement paste is in the range of 1.6 to 2.0 (calculated value is 1.74) [61,62]. Garbev et al. analyzed the incorporation limit of Ca in the C-S-H phase of nanocrystals by XRD and thermogravimetric analysis (TG) [63]. It was found that the Ca/Si ratio of the pure C-S-H phase was in the range of 2/3–5/4, and the structure of the C-S-H phase within this range can be well described by the defect tobermorite model. Chatterji directly measured the Ca/Si ratio of C-S-H fabricated by the hydration reaction of $C_3S$ using electron-beam microanalysis and compared the results with the indirect determination of CH extraction [64]. Both methods determined that the Ca/Si ratio of C-S-H was 1.6.

C-S-H gel has a pore structure with multi-dimension, and the diffusion of water and ions in the C-S-H gel determines the chemical and physical properties of cementitious materials, such as strength, creep and shrinkage [65]. Various experimental techniques have been utilized to study the properties of water confined to gel pores or the surface of C-S-H. Results from $^1$H-NMR tests indicate that water in C-S-H gel is divided into three types: chemically bound water with strong binding to C-S-H structure, physically bound water with deep adsorption near the surface of C-S-H and capillary water without binding and free diffusion in capillary pores [66,67]. Bordallo et al. quantitatively distinguished different water types in C-S-H gel by using the diffusion coefficient through quasi-elastic neutron scattering technology [68]. The unbound water molecules in the pore moved rapidly ($\sim 10^{-9}$ $m^2/s$), while the water molecules enclosed in the pore slowly diffused ($\sim 10^{-10}$ $m^2/s$). The change in water content in C-S-H gel will affect the molecular structure and performance of C-S-H. Besides, the water to silicate ratio (H/S) of C-S-H gels is sensitive to relative humidity, under saturation condition, the value of H/S is 4. When the relative humidity is 11%, the ratio of H/S is 2.1. Under dry conditions, the value of H/S is only 1.4 [69]. Density is another important parameter to be considered when the C-S-H model is proposed, and the density of the C-S-H gel is closely related to the water content in the gel pores. Jennings studied the density of C-S-H in different water-bearing states and found that the density of single-layer C-S-H excluding all evaporated water was 2.88 $g/cm^3$, and the density of saturated C-S-H without water adsorption on the surface was reduced to 2.602 $g/cm^3$ [70]. Brunauer et al. measured the density of dry C-S-H with a water hydrometer as 2.85 $g/cm^3$ [71]. Allen et al. measured the Ca/Si ratio of C-S-H at about 1.7 by small-angle neutron scattering and X-ray scattering techniques, and the statistical average molecular formula was $(CaO)_{1.7}(SiO_2)(H_2O)_{1.8}$, and the statistical average density was about 2.604 $g/cm^3$ [72]. This test method did not depend on the drying method, and the obtained density value may be very close to the real value of C-S-H gel density in the hardened cement paste. $^1$H NMR can characterize the nano-porosity of C-S-H in the prepared materials without drying. From the results of NMR, the density of C-S-H in the hardened cement paste varies with the degree of hydration ($\alpha$) and the water–cement ratio. When $\alpha$ increased from 0.4 to 0.9, the density of C-S-H without gel pore water decreased from 2.73 to 2.65 $g/cm^3$, the one including gel water increased from about 1.8 to 2.1 $g/cm^3$ [65].

Silicate anion is the basic unit of C-S-H gel. In the structure of silicate anion, the tetrahedral units $[SiO_4]$ are linked together by sharing oxygen atoms to form Si–O–Si bonds. Understanding the coordination of silicate anions is of great significance for the establishment of the C-S-H model. NMR is a powerful characterization method and provides valuable information for the structural analysis of silicate anions. Zanni et al. applied NMR to the detection of C-S-H structure, $^{29}$Si NMR gave the information of C-S-H with silicon-oxygen tetrahedron structure, $^1$H NMR distinguished the H element connected

with silicon atom or calcium atom, and [43]Ca NMR determined the location of calcium atom in the structure [73]. The location of the peak varied with the amount of bridging oxygen in a specific [SiO$_4$] unit. The [SiO$_4$] units can be classified by the position of the peaks in the spectrum as measured by [29]Si NMR. Theoretically, five types of Q$_n$ sites could be found, whose chemical displacement was Q$_0$ ($-70$ ppm), Q$_1$ ($-80$ ppm), Q$_2$ ($-88$ ppm), Q$_3$ ($-98$ ppm) and Q$_4$ ($-110$ ppm) [74]. From [29]Si MAS-NMR test spectrum, the C-S-H in the silicate chains was linear, Q$_0$, Q$_1$ and Q$_2$ were single silicon-oxygen tetrahedron (not connected to other silicon-oxygen tetrahedron), the end of the chain tetrahedron (only with a silicon oxygen tetrahedron) and central chain tetrahedron (connected to the two silicon-oxygen tetrahedron) in silicon oxygen tetrahedron, respectively. Figure 1 shows the position of the Si element in the C-S-H gel. Figure 2 shows the MAS-NMR spectrum of [29]Si. Silicon sites of Q$_1$ and Q$_2$ were dominant, Q$_3$ and Q$_4$ did not exist, Q$_0$ appeared in small amounts (representing the unhydrated clinker) [75]. Gautham et al. obtained the distribution information of silicon elements in C-S-H by NMR [76]. The test results showed that Q$_0$ $\approx$ 10%, Q$_1$ $\approx$ 67% and Q$_2$ $\approx$ 23% in C-S-H. Due to the interaction between silicate anions and Ca cations, the change in the Ca/Si ratio would lead to significant changes in silicate structure [77]. The analysis results of C-S-H samples with multiple Ca/Si ratios using X-ray photoelectron spectroscopy [78] and [29]Si NMR [79] showed that when the Ca/Si ratio remained unchanged, the average chain length increased with the hydration time. In addition, increasing the Ca/Si ratio will lead to a decrease in the polymerization degree of silicate chains. The depolymerization of Ca atoms is an important feature in the study of C-S-H skeleton structure.

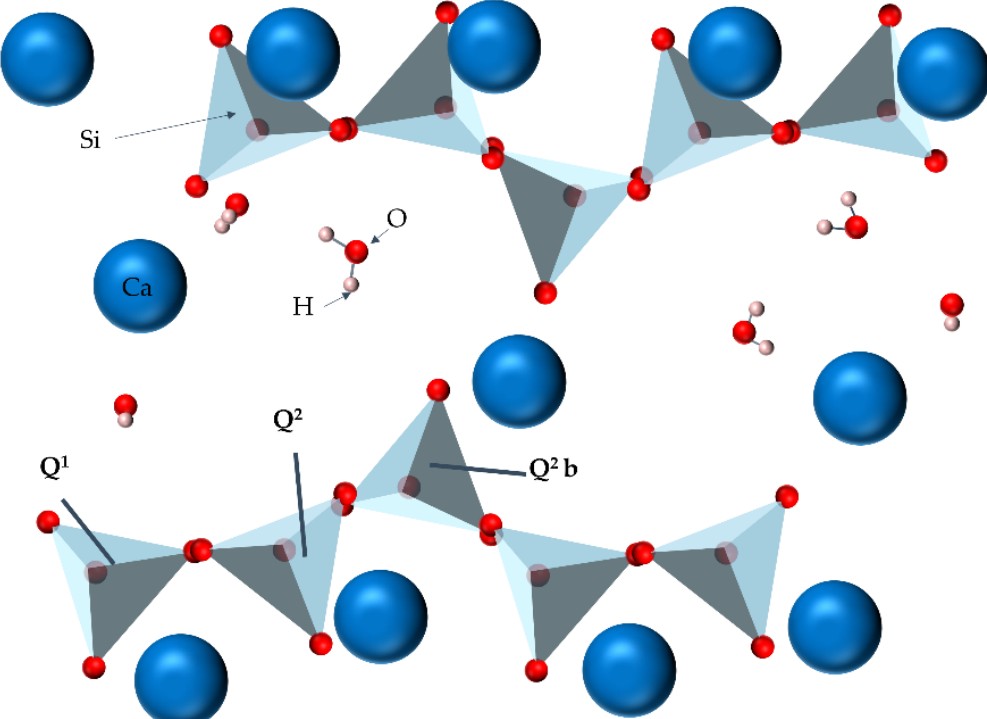

**Figure 1.** The bonding mode of Si element in C-S-H gel.

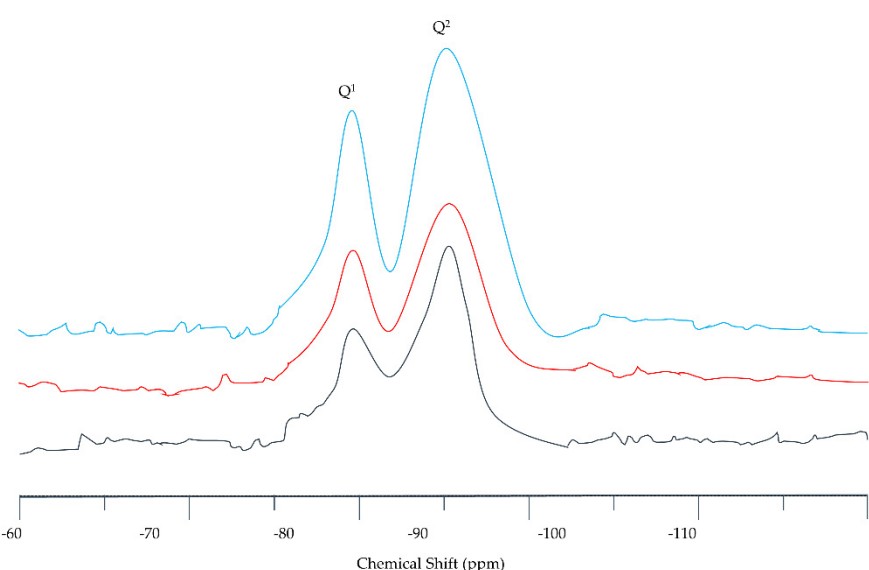

**Figure 2.** $^{29}$Si MAS-NMR spectrum of C-S-H.

### 3.2. C-S-H Theoretical Models

With the deepening investigation of C-S-H, the obtained molecular structure information of C-S-H is gradually enriched. In the past few decades, a variety of microstructure models have been proposed to describe the nanoscale structure of C-S-H (gels or globules) and/or the formation of gel pores. Several widely utilized models are introduced in this work. C-S-H gel that can be demonstrated by the first four models below has a porous structure. However, C-S-H globules have a smaller pore structure than C-S-H gels, so the model of C-S-H globules can be studied by molecular dynamics such as the fifth model.

- Powers–Brownyard model;

Powers and Brownyard proposed a C-S-H structure model similar to tobermorite based on the porosity test and water vapor adsorption test of the hardened cement paste [80,81]. In this model, the C-S-H gel is composed of particles with a layered structure, and the particles are composed of two or three layers of C-S-H, which can be folded into a fibrous form [82]. Randomly distributed layered structures connect with adjacent particles by surface forces with strong ionic covalent bonds. In this model, the water molecules in C-S-H gels are claimed to be divided into three categories: binding, gel and capillary water. The structure of C-S-H in this model will collapse during the drying, water is not allowed to re-enter. Therefore, the adsorption isotherm of the water is considered to be irreversible. This model is based on the adsorption experiment of water vapor. The model cannot directly follow the arrangement rules of the pores in C-S-H gel. Therefore, this model cannot deliver good explanations about creep and shrinkage.

- Feldman–Sereda model;

It is found that the structure of C-S-H is similar to the structure of layered crystals. In the model proposed by Feldman and Sereda (as showed in Figure 3 [83]), the thin slices that make up the C-S-H gel have irregular monolayer arrays of two to four molecules thick, which can randomly cluster together to form interlayer space, as demonstrated in Figure 3. Water is claimed to enter and leave the interlayer space of C-S-H at relatively low humidity; the physical adsorption of water bound to the C-S-H globules will affect the creep and shrinkage of the material. The bonding between layers is considered to be carried out by solid–solid contact. Weak van der Waals bonds and strong ionic covalent bonds are thought to play a connecting role in this contact. The interlaminar bonding is a special kind of chemical bonding and cannot be considered as the result of the interaction among the free surfaces of C-S-H layers.

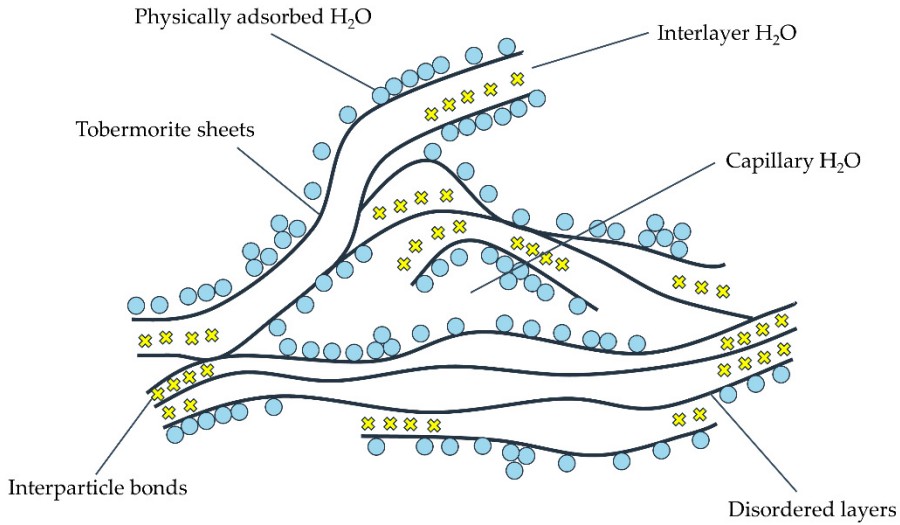

**Figure 3.** Feldman–Sereda model [83].

- Munich model;

In the Munich model (as illustrated in Figure 4), C-S-H gel is a three-dimensional network composed of amorphous colloidal gel particles (or say dry gels), which are bound together by van der Waals forces but still dominated by strong ionic covalent bonds [84]. Through results from a series of adsorption measurements, the role of water between the particles is emphasized in the model. It is explained that the creep is responsible for the weakening of bonds among C-S-H particles, rather than by the interlaminar binding force. Such a model well explains the behavior of hardened cement paste under different humidity conditions. Compared with the Feldman–Sereda model, the separation pressure in this model is assumed to occur in the high humidity region of the isotherm. Therefore, the model fails to reveal the phenomenon of creep or dry shrinkage under loading attributed to the change in C-S-H nano-pore. Similar to the Feldman–Sereda model, the model suggests that water vapor should interact with the C-S-H structure, making this model unsuitable for surface area measurements.

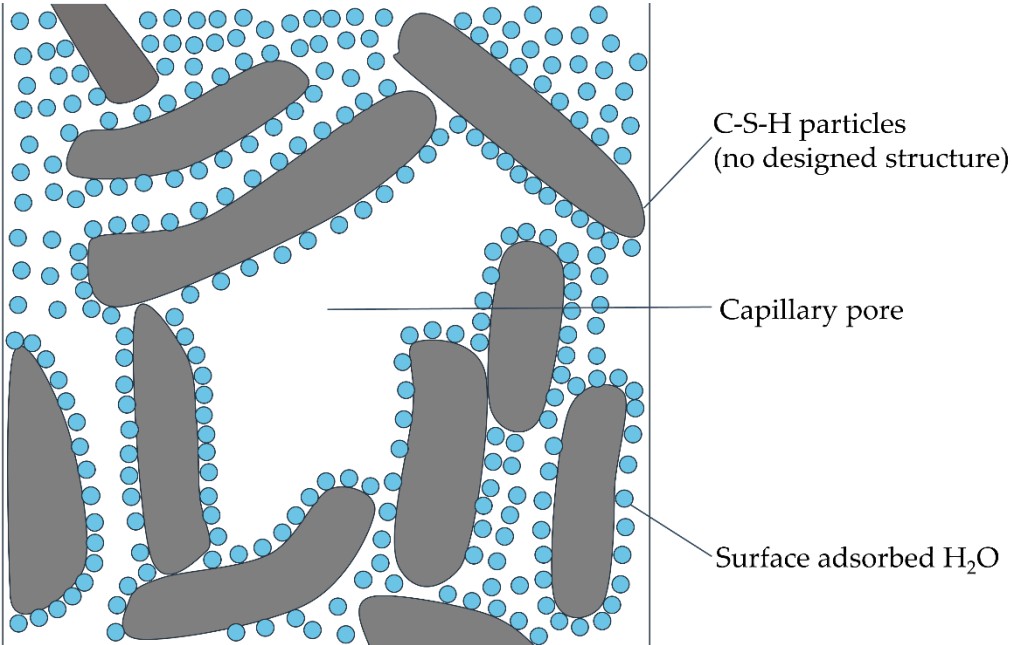

**Figure 4.** Munich model [84].

- Tobermorite–jennite model;

The tobermorite and jennite are both natural silicate minerals. The top view of the silicate chain connected to the calcium sheet in tobermorite structure is shown in Figure 5a. The side view of the double-layered tobermorite structure without interlaminar water molecules and interlaminar calcium ions is shown in Figure 5b. The top and bottom of the tobermorite silicate chain are connected to calcium, and the repeating units of the silicate chain contain at least three [SiO$_4$] tetrahedral units. Two tetrahedrons that share the oxygen in dome-shaped mode are called "paired tetrahedrons". The parasilicon-oxygen tetrahedron in both crystals locates on the main calc-oxygen layer, while the bridged silicon-oxygen tetrahedron connects the adjacent parasilicon-oxygen tetrahedron dimers, increasing the molecular chain length of the chain of Si-O. The Ca/Si ratio of jennite crystal is higher than that of tobermorite crystal, and there are Ca-OH groups on the main layer. According to different layer spacings, tobermorite crystal is referred to as 9, 11 and 14 Å mineral [85]. Tobermorite minerals have different interlamellar spacings, which usually contain water molecules and calcium ions (or other cations) [86]. Tobermorite of 11 and 14 Å are proved to be monoclinal crystals, while 9 Å tobermorite and jennite are triclinic crystal structure [87]. Jennite (as illustrated in Figure 6) is another crystalline mineral whose structure has a high similarity to the structure of C-S-H gel.

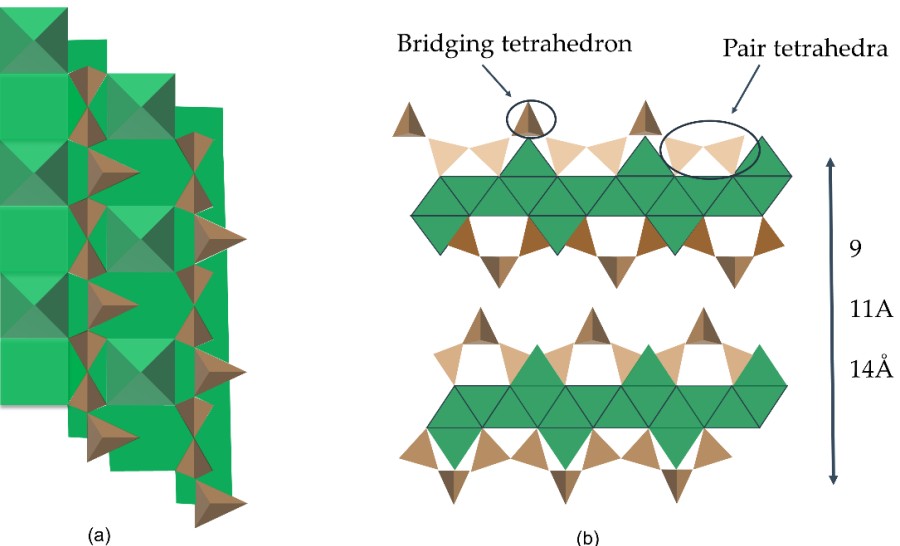

(a)   (b)

**Figure 5.** Molecular structure models of tobermorite [86,88]. (**a**): The top view of the silicate chain connected to the calcium sheet in tobermorite structure; (**b**): The side view of the double-layered tobermorite structure without in-terlaminar water molecules and interlaminar calcium ions.

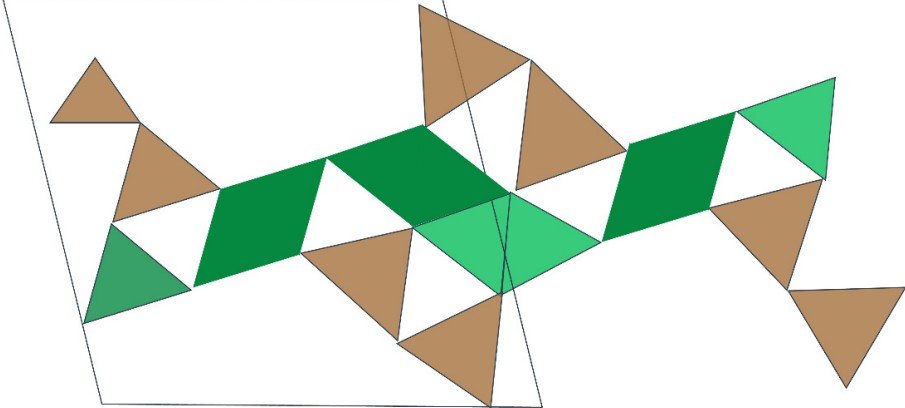

**Figure 6.** Molecular structure models of jennite [86].

Taylor tried to divide C-S-H into two significant forms by experimental parameters such as Ca/Si and density of C-S-H: C-S-H (I) and C-S-H (II). C-S-H (I) was close to 14 Å tobermorite when Ca/Si < 1.5; C-S-H (II) was likely to jennite when Ca/Si > 1.5. The tobermorite–jennite model proposed by Taylor considered that C-S-H was a disordered layered structure, most of the layers were similar to the structure of 14 Å tobermorite, and a small number of layers were similar to the structure of jennite with defects [89].

In recent years, a large number of studies have shown that tobermorite, rankinite has high similarity in structure with C-S-H [90–93]. Therefore, many C-S-H models based on tobermorite or tobermorite/rankinite combinations were proposed. Skinner et al. prepared C-S-H (I) crystals in hydrated $C_3S$ paste. Synchrotron X-ray scattering measurements showed that the nanocrystal size of C-S-H (I) crystals was 3.5 nm, similar to the 11 Å tobermorite crystal structure with size enlarged [94]. Bonaccorsi et al. also considered that jennite had a high degree of similarity with C-S-H (II) [85,87].

A number of models for the nanostructure of C-S-H are summarized and compared, and it is shown that the value of the structures of 1.4 nm tobermorite and jennite for visualizing the nanostructural elements present in the models is demonstrated [95]. Richardson et al. proposed the tobermorite/jennite model and tobermorite/CH model, in which the effect of CH and possible protonation were considered, and the model was highly consistent with their experimental results by X-ray microanalysis [95–97]. Grutzeck investigated the formation of C-S-H that was firstly a rapid equilibrium process, followed by a slow diffusion phase transition process. If the calcium concentration was lower than the level required for the stable phase, C-S-H would undergo a phase change, forming tobermorite and jennite structure C-S-H nano-fragments [98]. Chen et al. compared C-S-H gels in $C_3S$ hydrated paste and C-S-H synthesized by the chemical method through NMR and XRD. According to their results, when CH solution was saturated, the structure of C-S-H gels can be described by 14 Å tobermorite, jennite or a combination model of both.

- Simulated model;

Although the models of C-S-H gels discussed above are based on some of their structural characteristics, the structures described by these models do not completely match some new findings from experimental tests. In recent years, based on advances in computational science, some researchers have built models of C-S-H globules using molecular dynamics simulation. The molecular simulation model takes the structural information of C-S-H globules obtained from some experimental results systematically into account. The similarity between the model structure and C-S-H globules structure is greatly improved. Compared with theoretical models, computational models can perform some operations that are impossible or extremely time-consuming in reality at the atomic level and can predict additional basic properties of C-S-H. Some of the known computational models are based on the structures of tobermorite or jennite nanocrystals. Gmira et al. established a model based on the crystal structure of tobermorite and studied the bond of particles in C-S-H by using atomic potential energy minimization and quantum chemical simulation [99]. Pellenq et al. conducted simulation in atomic simulation using molecular dynamics and energy minimization technology and showed that the interlayer cohesion of C-S-H was caused by ionic covalent bonds. The elastic modulus calculated by them was in good consistence with the test data of the atomic force microscope [100]. Then, their team built a C-S-H calculation model with the Ca/Si ratio of 1.7 based on 11 Å tobermorite and added short silicate chains distributed as monomers, dimers and pentamers. The analysis showed that C-S-H gel had both the short-range ordered characteristics similar to glass and tobermorite [101]. In the molecular dynamics work by Dolado et al., when the Ca/Si ratio was 0.9–1.3, the model of C-S-H globules were similar to the structure of 14 Å tobermorite. When Ca/Si ratio was 1.5–2.3, the model approximated the structure to be jennite-like [102]. Figure 7 is one of C-S-H structure simulated by molecular dynamics. Table 2 displays several C-S-H models.

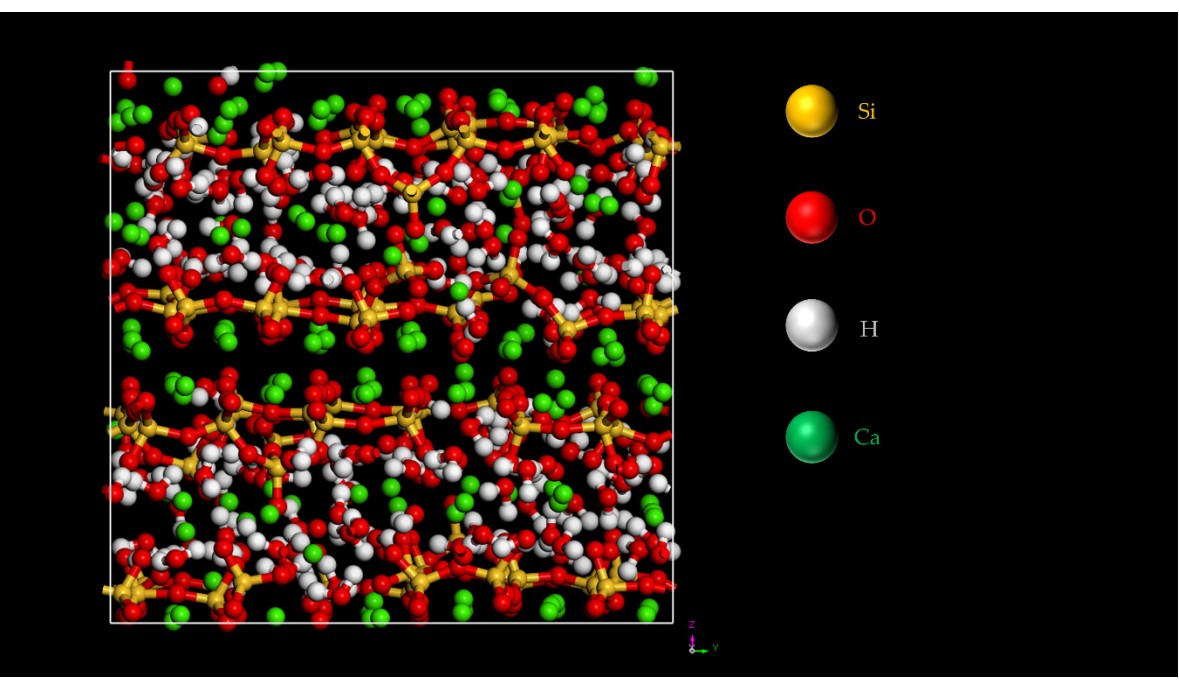

**Figure 7.** One of the C-S-H structures simulated by molecular dynamics [14].

**Table 2.** Several C-S-H models.

| Models | Foundation | Features | Disadvantages |
|---|---|---|---|
| Powers–Brownyard model [80–82] | Porosity test, water vapor adsorption test | Layered structure, collapse when dry, water not allowed to re-enter | Unable to follow the arrangement rules of the pores in the C-S-H gel, unable to explain the creep and shrinkage |
| Feldman–Sereda model [83] | Based on the test results of nitrogen adsorption isotherms | Solid contact, weak van der Waals bonds, strong ionic covalent bonds | Based on few features of C-S-H |
| Munich model [84] | Based on adsorption measurement | Van der Waals bonds, strong ionic covalent bonds | Fails to explain the creep or drying shrinkage, unable to describe pore structure |
| Tobermorite–jennite model [85–87,89–98] | Natural silicate pore structure | Highly similar to C-S-H structure | Pore structure cannot be explained clearly |
| Molecular dynamics model [99–102] | Molecular dynamics | Atomic level performs operations; predict structural characteristics | Need experimentation to verify its correctness and applicability |

## 4. Fractality of C-S-H

According to Mandelbrot [103], the fractality characterizes a self-similar structural pattern on a certain scale, which can be characterized by the so-called fractal dimension. Fractal dimension allows for the detection fractal patterns of an object over a range of scales. Fractal dimension is usually a non-integer value. The concept of fractal dimension is widely utilized to describe the irregular behavior and disorganized phenomenon [103]. Later, it was used to study the non-uniform surface structure of solid materials, thus providing new tools and methods for cement-based material analysis [104]. According to fractal theory, the surface of C-S-H gel has fractal characteristics on a molecular scale, which means irregular defects or degree of the surface are similar at different spatial scales [105]. Fractal surface can be described by fractal dimension (value between 2 and 3), which reflects the rough or irregular degree of the surface. When the value equals 2, it shows that the surface is smooth and regular, if the value is nearly 3, it shows that the surface is rough and irregular.

The fractal dimension cannot be evaluated directly, but it can be determined indirectly by some advanced testing methods, such as nitrogen ($N_2$) adsorption (NAD), NMR, small angle X-ray scattering (SAXS) and mercury intrusion porosimetry (MIP), in which $N_2$ adsorption method is relatively widely utilized. $N_2$ adsorption is based on the nitrogen isotherm of adsorption and desorption; a specific surface area of specimen can be calculated utilizing the Brunauer–Emmett–Teller formula. The micropore analysis method, Dubinin–Radushkevich method and Horvath–Kawazoe method, which are suitable for Barret–Joyner–Hallenda as well as micro-pore analysis (diameter below 2 nm), which is suitable for meso-pore analysis [106] (diameter is between 2~50 nm), are utilized to calculate other pore structure parameters. Fractal dimension can be calculated by the Frenkel–Halsey-Hill (FHH) model and Neimark model. The FHH model is based on multilayer adsorption, and the fractal dimension can be obtained from nitrogen adsorption isotherms. Neimark et al. [58] utilized the thermodynamic method to establish the relational expression of the fractal dimension of the solid surface. When the adsorbate undergoes capillary condensation, the liquid film replicates the surface characteristics of the sample. The fractal size is based on the area of the liquid film and the average radius of curvature of the liquid film. Sahouli et al. proved the equivalence of FHH and Neimark methods [59]. For instance, Kriechbaum et al. [107] utilized the SAXS to find that there was an obvious increase in volume or mass fractal dimension ($D_m$) of cement paste with curing age, but Tang et al. [108] utilized the NAD to suggest that neither volume nor surface fractal dimension ($D_s$) showed obvious mutual regularity.

Various methods have been developed to evaluate the fractal structures of random surfaces. Generally, they are used in both the Hausdorff and box dimensions, which can be defined on any metric space. Thus, while the definition of former is based on measurement (from a theoretical approach), the latter becomes easier to be empirically calculated or estimated (from the viewpoint of applications). In this way, most empirical applications involving fractal dimension have been carried out in the context of Euclidean spaces through the box dimension [109,110]. Due to the features of ease of utilized simple algorithms and relatively high accuracy, the box-counting method may be one of the most widely utilized methods to evaluate the fractal dimension of complex surfaces [111–113]. Some backgrounds of the box-counting method for calculating fractal dimension of a structure can be found in Refs [114,115]. For a given structure, its box dimension could be calculated by a regression that compares the number of boxes (more generally, covers) of a given length (resp., diameter) over a range of scales, the box-counting dimension can be expressed as a mesh of squares [114–117]. The fractal dimension measures the rate at which the number of boxes of a given length increases as such a length (the scale) decreases. Practically, the fractal dimension is highly dependent on the algorithm by which the numbers of objects in a certain measure (e.g., length, surface area, volume or mass) are estimated [103,117–119]. Lü et al. found that the errors of the estimations are inevitable, especially some imagination-based models with a threshold operation before calculating fractal dimension [120]. However, some authors utilized a novel approach to accurately calculate the fractal dimension (i.e., both box and Hausdorff dimension) of objects contained in a Euclidean space, which allows for the calculations being carried out in the real line by taking into account the preimage of a space-filling curve, such as Hilbert's one (take a look at the references below) [121,122]. For instance, Fernández et al. provided several approaches, such as counting the number of $2^{-n}$ cubes—involving a distance function—and a discretized Hausdorff dimension, to generalize the box dimension in the context of fractal structures, which would allow us to calculate the fractal dimension in a wider range of spaces and situations [123].

Surface roughness refers to the small spacing and small peaks and valleys on the surface of the material. The wave distance between its two wave crests or two wave troughs is below 1 mm. Surface roughness is generally formed by different preparation methods of materials, such as different parameters in the preparation process, different uses of raw materials and some environmental factors during storage. For instance, C-S-H gel with

different Ca/Si ratios has different surface roughness. The surface roughness of C-S-H gel synthesized by different methods such as the sol-gel method and hydration of $C_3S/C_2S$ is also different. The depth, density, shape and texture of the traces on the surface are also different.

Shen et al. [124] prepared concrete with an ultra-smooth surface. The surface of this concrete was covered with a nano-membrane of C-S-H nano-particles, the size of which was approximately 10 nm. The surface roughness of ultra-smooth concrete could be analyzed by atomic force microscopy (AFM). The surface roughness was calculated based on finding the median surface level of the image, then evaluating the standard deviation. The result showed that the surface roughness of ultra-smooth concrete was much smoother than the polished concrete [124]. The surface roughness of some areas was composed of compacted and packed nano-particles, with almost no defects around 1–4 nm. Other areas were composed of loosely packed particles between 4–10 nm. The surface roughness of polished cement paste was about 13–33 nm, which was much higher than that of the ultra-smooth concrete [125].

In addition to MIP and NAD, low-temperature differential scanning calorimetry (LT-DSC) and small-angle neutron scattering (SANS) can be utilized to evaluate fractal dimension indirectly. Some experimental results from these methods are compared to illustrate the advantages and disadvantages of such methods [126].

Self-similarity is a typical characteristic of fractality [127–130]. Some properties (thermal conductivity, permeability, diffusivity, fluidity and strength) of cement-based materials are closely related to the fractal feature of microstructure. Apart from porosity, the thermal conductivity and permeability are sensitive to fractal dimensions associated with pore volume and tortuosity. The compressive strength of cement-based materials increases with the increase in the fractal dimension, while the fluidity of cement paste is negatively related to the fractal dimension associated with particle size distribution of cement [126]. The study of fractality of C-S-H is beneficial to explore the structure and mechanical performance of C-S-H from a microscopic perspective. Shen et al. utilized atomic force microscopy (AFM) to find out that the C-S-H gel had self-similar features ranging from a scanning scale of $20 \times 20$ μm to $300 \times 300$ nm, and the features of large-scale C-S-H were very similar to those on a small scale. It can be concluded that C-S-H gels were composed of some basic spherical balls aggregated into larger spheres [131].

Table 3 summarizes the experimental methods for the pore structure of cement-based materials. The choice of the fractal analysis method for cement-based materials mainly depends on the purpose of the investigation. For example, NAD and LT-DSC can be regarded as effective measures to detect the fractal features of C-S-H gels.

Some authors utilized MIP to study the surface fractal dimension of C-S-H gel. Yang et al. found that the fractal dimension of gel pores was almost the highest in the entire pore size range. The high value of $D_s$ reflected the extremely disordered and complex nature of the C-S-H gel, especially over the mesopore range 6–25 nm [132]. The C-S-H gel reverted to regular morphology as it expanded in the large capillary pores. Subsequently, as the swelling of the C-S-H gel became less active due to the presence of unhydrated cement and powdered blast furnace slag (GGBFS), $D_s$ increased. The C-S-H gels usually had a fibrillary one-dimensional shape, while the ones had a two-dimensional corrugated foil shape in the presence of GGBFS [133–135].

Zeng et al. utilized the Neimark model with cylindrical assumption and Zhang model without geometry assumption to study the pore fractal dimension [136]. In both models, the logarithm plots exhibited the scale-dependent fractal properties, and three distinct fractal regions (I, II, III) were identified for the pore structures. According to the Neimark model, the fractal dimension of pores was estimated to be 2.592–2.965 and, according to Zhang's model, was 2.487–2.695. Region III might be related to the interlayer or intergranular pores of C-S-H gels [136]. According to the work by Jennings et al. [137,138], these C-S-H gels formed a globule with an inherent gap of about 2.2 nm, and then, these globules assembled together to form a C-S-H solid, with an inherent gap of about 5.6 nm. Using the

conceptual model of spherical flocs of C-S-H hydrate [20], they concluded that C-S-H gel was composed of spherical flocs with a fractal dimension of 2.67 [70,139].

**Table 3.** Summary of experimental methods for pore structure of cement-based materials [126].

| Methods | Principle | Limitations | Measuring Range |
|---|---|---|---|
| MIP | The fluid and air are pulled out of specimen, and then mercury is intruded into pores of specimen by pressure. Measure the volume of mercury entering the specimen pore under pressure, and porosity can be computed by Washburn equation | 1. C-S-H gel pores are different from the real cylindrical shape required by the Washburn equation; 2. The pressure that causes mercury to enter the pores may damage the pores; 3. Unconnected pores are difficult to be detected. | Several nm–hundreds of µm |
| NAD | The adsorption isotherms are determined and pore size distribution is computed based on the Kelvin equation | 1. The micro-pores are not determined; 2. Nitrogen molecules are larger than water molecules and cannot enter very fine pores | $0.5-100$ nm |
| LT-DSC | Specimen is saturated with water. Freezing–melting process is required. The registered heat flux is measured by the Gibbs–Thomson equation | 1. Results vary fairly; 2. Limited pore range; 3. Freezing–melting process may affect pore structure | 2–1 µm |
| SAXS and SANS | Scattering of X-rays or neutrons occurs because of the scattering contrast among various components. X-rays probe the fluctuations of the electronic density, whereas neutrons probe the fluctuations of the nuclear scattering cross section. | 1. The relation between scattering data and pore structure is not well established; 2. Expensive to measure | 1–20 µm |

$D_m$ is used as a function of saturation, hydration time and degradation [107,140–142]. SAS technology is the preferred method for studying $D_m$. Francesca et al. utilized SANS/SAXS to study the pure C-S-H phase and found that its $D_m$ was 2.8, dropping to 2.6 when superplasticizers was incorporated. The value of $D_m$ extracted by LT-DSC was 66% lower than that by SANS/SAXS. Another study used the SANS test and found that the size of the C-S-H fractal increased with the increase in W/C, because the structure of the C-S-H gel became compact [143].

Table 4 is the comparison of $D_s$ and $D_m$ of C-S-H gels from the previous literature. It is noted from this table that it is difficult for C-S-H gels to obtain recognized fractal dimensions. Differences in results of different methods are sometimes attributed to the inherent assumptions used.

**Table 4.** Comparison of $D_s$ and $D_m$ of C-S-H gels from previous literature.

| Fractal Dimension | Test Method/Condition | Disadvantages |
|---|---|---|
| From FHH model and Neimark's model [144] | Nitrogen adsorption, oven dry | Fractal dimension does not have a direct relationship with other pore structure parameters (such as pore volume or average diameter) |
| From Neffati's model [143] | Low temperature differential scanning calorimetry | Within 7–70 nm, no obvious relation among fractal dimensions, hydration age and superplasticizers content |

## 5. Mechanical Properties of C-S-H

### 5.1. Mechanical Experiments

Mechanical properties are the basis for the extensive application of cement-based materials [145]. In terms of experimental research, the current mechanical properties test of C-S-H is mainly based on cement paste or C-S-H powder as the research object, and the nano-indentation test is the main technical means to evaluate the mechanical properties of

C-S-H. When the C-S-H gel is selected as the research object in cement pastes, its identification is particularly critical. The testing system of nano-indentation combined with electron microscope and element analysis module is the main means to study the mechanical properties of C-S-H in cement pastes. Zhu et al. established a mechanical testing-constitutive phase-elemental analysis integrated system by combining nano-indentation apparatus, SEM and EDS. Low-density C-S-H (LD-CSH) and high-density C-S-H (HD-CSH) were identified. The hardness ($H$) and elastic modulus ($E$) of two kinds of C-S-H gels were obtained by nano-indentations: $H_{LD-CSH} = 0.73 \pm 0.15$ GPa, $H_{HD-CSH} = 1.27 \pm 0.18$ GPa; $E_{LD-CSH} = 23.4 \pm 3.4$ GPa, $E_{HD-CSH} = 31.4 \pm 2.1$ GPa [146]. With the help of same methodology, Constantinides et al. found through a nano-indentation test that the $E_{LD-CSH}$ was about 22 GPa, while $E_{HD-CSH}$ was about 29 GPa [147]. Hu et al. found that the Ca/Si ratio of C-S-H gel in hydration products decreased with the increase in W/C ratio of cement pastes. The results of nano-indentation on C-S-H gels with different Ca/Si ratios showed that their mechanical properties were affected by the porosity and its volume fraction. Specifically, a high W/C ratio would lead to a low $E$ of inner and outer product C-S-H gels [148]. Sebastiani et al. utilized the same methodology to point out that the transformation from low density to high (or ultra-high) density improved the mechanical strength of C-S-H [149].

Due to the complexity and diversity of cement hydrated products, the exact selection of C-S-H gels is very difficult, and the presences of other hydrated products and pores strongly affect the accuracy of the nano-indentation test [148,150]. With the help of chemical fabrication methods, the purity of C-S-H is greatly improved, which is beneficial to the study of the influence of chemical components on its mechanical properties. Feldman found that interlayer water had an important impact on the mechanical properties of C-S-H, and dehydration would make $E$ of C-S-H reduced [151]. Plassard et al. analyzed the mechanical properties of synthetic C-S-H by AFM. The results demonstrated that $E$ of C-S-H increased with the increase in Ca/Si ratio [152]. However, some studies have reached the opposite conclusion, namely, $H$ and $E$ increase with the decrease in the Ca/Si ratio of C-S-H [150,153]. Beaudoin et al. also found that $E$ of C-S-H appeared to be independent of Ca/Si ratio and the degree of silicate polymerization [153,154]. Some studies believed that the pores among the synthetic C-S-H powders would affect the results of microscopic mechanical test [147,153,155,156], which may lead to discrepancies of mechanical results of C-S-H. Table 5 shows mechanical experiments of C-S-H.

**Table 5.** Mechanical experiment of C-S-H.

| Methods | Findings | Disadvantages |
|---|---|---|
| SEM,EDS [146] | LD-CSH and HD-CSH are identified | Cement products are diverse and complex; low accuracy of nano-indentation test; pores in the concrete structure; purity of sample optimized |
| Nano-indentation [148] | when W/C ratio in cement paste increases, the Ca/Si ratio decreases | |
| Nano-indentation [149] | $C_3S$ products are tested, transformation from LD to HD improves the mechanical strength of C-S-H gels | |
| Nano-indentation [151] | Interlayer water affects the mechanical properties of C-S-H, dehydration causes $E$ to decrease | Mechanism behind has not been revealed yet |
| AFM [147] | $E_{LD-CSH}$ about 22 GPa, $H_{HD-CSH}$ about 29 GPa | Findings cannot reach a consensus, Ca/Si ratio of C-S-H decreases, $H$ and $E$ increase |
| AFM [152] | The $E$ of C-S-H increases with the increase in the Ca/Si ratio | |
| AFM [150,153–156] | The $E$ of C-S-H is independent of Ca/Si ratio and silicate polymerization degree | |

### 5.2. Mechanical Simulation

In the simulation methodology, the first principles and molecular dynamics simulation are common means. It is difficult to study materials at the nanoscale through some physical experiments; nevertheless, the computer simulation may be used to study materials at molecular or even atomic dimension. The molecular composition can be controlled accurately, which is very helpful to study the mechanical properties (such as elastic modulus, hardness, fracture toughness, etc.) of C-S-H.

Bauchy et al. used molecular dynamics to simulate the fracture toughness of C-S-H at the atomic dimension [157], and the results showed that at this dimension, the fracture of C-S-H was manifested as a ductile fracture. Shahsavari et al. evaluated the mechanical properties of C-S-H with different crystal structures by using the first principle based on the structure of tobermorite and jennite [86]. The investigated class of materials and results are relevant to chemically complex hydrated oxides, such as layered C-S-H, the binding phase of all concrete materials and the principle source of their strength and stiffness. Zhang et al. made use of the CSH-FF force field and found that the strength and fracture toughness of C-S-H increased after CH entered the C-S-H layer, which was caused by the strong hydrogen bond formed between CH and C-S-H [158]. Hou et al. established a hierarchical C-S-H model at the molecular level based on the crystal structure of tobermorite. Through the molecular dynamics calculation, it was found that the $E$ of C-S-H was about 40–50 GPa in X and Y directions and about 25 GPa in the Z direction under uniaxial tensile load [159]. Manzano et al. studied the mechanical properties of C-S-H through lattice dynamics simulation. The results showed that the volume, shear stress and elastic modulus of C-S-H decreased slightly with the increase in Ca/Si ratio, and the decrease trend was obvious with the increase in the ratio of $H_2O$/Ca [160]. Through the molecular dynamics simulation, Qomi et al. found that there was a significant decrease in the average indentation modulus with increasing Ca/Si ratio [161]. It is not surprising that as Ca/Si ratio increases the calcium-silicate layers become defective, and as a consequence, mechanical stiffness and anisotropy decrease. A similar trend is found for the hardness $H$.

Mechanical strength is often the index to evaluate the durability of cement-based materials to some degree. Brown et al. combined the nano-indentation apparatus, SEM and EDS, analyzed the mechanical response of each nano-indentation position with the chemical composition and measured $E$ of the original hardened cement paste to be $31.1 \pm 0.9$ GPa. $E$ of the C-S-H phase in decalcified cement paste decreased to be $13.7 \pm 1.7$ GPa [162]. Ashraf et al. conducted a comparative study on the microstructural phase of $C_3S$ hydration products before and after carbonization and found that the hydration and carbonization paste of $C_3S$ contained C-S-H, CH and $CaCO_3$, but the Ca/Si ratio of C-S-H phase in carbonized specimens was lower than that of hydration specimens. The nano-indentation test showed that $E$ of C-S-H, $CaCO_3$ and $C_3S$ grains decreased by about $39 \pm 5$ GPa, $59 \pm 6$ GPa and $110 \pm 30$ GPa [163]. Table 6 shows the mechanical simulation of C-S-H, and Table 7 shows the mechanical parameters of C-S-H.

**Table 6.** Mechanical simulation of C-S-H.

| Researchers | Principle | Findings | Possible Improvements |
|---|---|---|---|
| Bauchy et al. [157] | Molecular dynamics | C-S-H fracture appears as ductile fracture | Verified by various experiments |
| Qomi et al. [161] | Molecular dynamics | the indentation modulus and hardness of C-S-H decrease with the increase in Ca/Si ratio | |
| Hou et al. [159] | Molecular dynamics | A layered C-S-H model at the molecular level is constructed | Structure of tobermorite, jennite and C-S-H are not completely consistent |
| Shahsavari et al. [86] | First principles | Mechanical properties of C-S-H with different crystal structures are evaluated | |
| Zhang et al. [158] | Molecular dynamics | After CH enters the C-S-H interlayer, its strength and fracture toughness increase; a strong hydrogen bond is formed | Consider a variety of positions for analysis |
| Manzano et al. [160] | Lattice dynamic simulations | Ca/Si ratio increases, the volume, shear stress and $E$ of C-S-H all decrease | C-S-H is an amorphous structure |

**Table 7.** Mechanical parameters of C-S-H.

| Mechanical Parameters | Ref. |
|---|---|
| $H_{\text{LD-CSH}} = 0.73 \pm 0.15$ GPa, $H_{\text{HD-CSH}} = 1.27 \pm 0.18$ GPa; $E_{\text{LD-CSH}} = 23.4 \pm 3.4$ GPa, $E_{\text{HD-CSH}} = 31.4 \pm 2.1$ GPa | [146] |
| $E_{\text{LD-CSH}} = 22$ GPa, $E_{\text{HD-CSH}} = 29$ GPa | [147] |
| $E_{\text{X,Y}} = 40$–$50$ GPa, $E_{\text{Z}} = 25$ GPa | [159] |
| Original hardened cement $E = 31.1 \pm 0.9$ GPa; decreased to be $13.7 \pm 1.7$ GPa | [162] |
| $E$ of C-S-H, $CaCO_3$ and $C_3S$ about $39 \pm 5$ GPa, $59 \pm 6$ GPa and $110 \pm 30$ GPa | [163] |

## 6. Durability of C-S-H

### 6.1. C-S-H Carbonization

Cement-based materials are served in a carbon dioxide environment for a long time. The hydration products of cement chemically react with carbon dioxide, which will cause changes in the composition and structure of the materials. Macroscopically, the carbonization of C-S-H is manifested as changes in a series of volume and strength of hydration products. At present, a large number of studies on the carbonization of cement-based materials believe that carbon dioxide, as an acidic medium, will react with CH in the hydrating paste to form crystalline $CaCO_3$, which possibly reduces the pH value of the hardened cement paste and causes corrosion of the steel bars. Therefore, most of researchers focus on the carbonization of CH. C-S-H gel, as the main hydration product of Portland cement paste, has not received widespread attention on its carbonization mechanism.

The carbonization of C-S-H is a kind of chemical reaction, which is particularly important for the analysis of the reaction process and products. Sevelsted et al. utilized [29]Si NMR to reveal that C-S-H decomposition caused by carbonation involved two steps, and the decomposition rate decreased with the increase in Ca/Si ratio [164]. The first step was the gradual removal of calcium from the interlayer and defect of the C-S-H silicate chain until the ratio of Ca/Si = 0.67. The calcium in the main layer was consumed in the second step, resulting in the final cleavage of C-S-H and the formation of amorphous silicon phases consisting of $Q_3$ and $Q_4$ silicate tetrahedra. Morandeau et al. explored the change in C-S-H atomic structure caused by carbonation by in situ X-ray full scattering measurement and the distribution function [165]. The results showed that within 27 min of the initial interaction between C-S-H and carbon dioxide, a large amount of spherical aragonite and calcite were generated on the surface of C-S-H, followed by the gradual conversion of spherical aragonite into calcite. Besides, the residual calcium in the amorphous decalcified gel may

exist in the form of the amorphous calcium carbonate phase. Chang et al. accelerated carbonization of C-S-H with a Ca/Si ratio of 1.5 for 2 h at the $CO_2$ pressure of 0.2 MPa and quantitatively characterized the carbonization products and carbonization degree of C-S-H by quantitative X-ray diffraction, mass increment method and TG/MS [166]. C-S-H with $CaO/SiO_2$ ratio of 1.5 target mixture was synthesized. From the TG analysis, $C_{1.39}SH_{1.14}$ with 93.6% purity was obtained. After accelerated carbonation at 0.2 MPa $CO_2$ pressure for 2 h, the carbonation degree from mass gain method was 71.5%, and from TG/MS analysis was 78.0; the final C-S-H was $C_{0.28}SH_{0.15}$.

The composition of C-S-H is complex, and its elemental composition and proportion have an important influence on carbonization behaviors. Black et al. combined NMR and XRD to conduct carbonization experiments on C-S-H under natural conditions for 6 months. The results showed that the amorphous calcium carbonate hydrate was formed within a few minutes after C-S-H was exposed to air. With the extension of carbonization time, the carbonization products of C-S-H were dominated by Vaterite when the Ca/Si ratio $\geq 0.67$. The carbonization products of C-S-H were dominated by Aragonite when the Ca/Si ratio $\leq 0.50$. When the Ca/Si ratio of the C-S-H phase was 0.67–0.75, its carbonation resistance was the largest, and its structure was still dominated by $Q_2$ silicate even after 6 months of storage [167].

C-S-H in hardened cement paste is usually a hybrid material with multiple chemical elements introduced. It is of great practical significance to understand the influence of other chemical elements on C-S-H carbonization. Sevelsted et al. used $^{27}Al$ NMR to characterize C-A-S-H with Ca/Si ratios of 1.0 and 1.5 and found that all-aluminum sites related to C-S-H were consumed in the carbonization process and were mainly incorporated into amorphous silicon phase in the form of tetrahedral $Al(-OSi)_4$ units, and a small amount of pentagonal Al sites were also found in the silicon phase [164]. Morandeau et al. found that cement paste with high magnesium could form stable amorphous calcium carbonate-containing magnesium, prevent the removal of additional calcium in C-A-S-H gel and, thus, eliminate the carbonization reaction [168]. Li et al. studied the effects of Al and Mg on C-S-H carbonization by XRD, NMR and thermogravimetric analysis (TGA). The results showed that the addition of Al increased the content of C-S-H bridging silicate tetrahedron ($Q_2$). When C-S-H was mixed with magnesium, the cross-linking structure ($Q_3$) appeared. Compared with C-S-H, C-M-S-H had more polymerization units, stronger bond strength and better carbonization resistance [169].

The carbonization research under the state of C-S-H in the hardened cement paste has important guiding significance for the production and curing of cement-based materials. As early as 1977, Bensted applied Raman spectrum technique to analyze the carbonation performance of cement hydration products [170]. Castellote et al. applied neutron diffraction to conduct comparative studies on the carbonization of each phase in cement paste. The results showed that AFt, CH and C-S-H gel in the hardened cement paste decreased, while the content of calcite increased gradually. In terms of the carbonization reaction rate, AFt had the fastest reaction rate, followed by C-S-H gels, while CH had the slowest reaction rate [171]. Morandeau et al. conducted a comparative study on carbonization of CH and C-S-H in cement paste and mortar. The results demonstrated that CH and C-S-H carbonization occurred at the same time, and the initial carbonization rate was very close to each other. However, the carbonization rate of CH decreased significantly in the late age. When C-S-H was still carbonized, the carbonization of CH slowed down and even stopped. In addition, the carbonization could significantly reduce the porosity of the material. Decalcification occurred when C-S-H was carbonized, resulting in a decrease in the Ca/Si ratio of C-S-H. The release of calcium ions into gels' pores resulted in a decrease in the molar volume of C-S-H [172]. Castellote et al. performed phase analysis on hardened cement paste accelerated by carbonation with different concentrations of $CO_2$. The results showed that carbonization led to the gradual polymerization of C-S-H and the formation of Ca modified silica gel and $CaCO_3$. The carbonization reactions of C-S-H and CH occurred simultaneously, and the polymerization of C-S-H strengthened with the increase in $CO_2$

concentration after the carbonization [173]. Ibanez et al. utilized micro Raman spectroscopy to study the hydration and carbonization process of $C_3S$ and $C_2S$ and found that the crystallization degree of hydration products of both minerals was significantly reduced, and heterogeneous calcite, aragonite and pellet aragonite were formed in carbonized specimens. The carbonation in $C_3S$ hardened cement paste was limited to the depth of 500~1000 μm from the surface, while the carbonation depth of $C_2S$ paste was far larger than that of $C_3S$ [174].

A large number of studies have proved that environmental conditions have a great influence on the carbonization of C-S-H. Ceukelaire et al. studied the influence of relative humidity on C-S-H carbonization and found that the carbonization rate was the largest when the ambient humidity of C-S-H was 50−70% RH [175]. Table 8 shows the comparison of the works on the carbonization of C-S-H by different researchers.

**Table 8.** Comparing the works on the carbonization of C-S-H made by different researchers.

| Researchers | Methods | Findings |
|---|---|---|
| Sevelsted et al. [164] | $^{29}$Si NMR | The decomposition rate decreased with the increase in the Ca/Si ratio |
| Morandeau et al. [168] | X-ray scattering | C-S-H reacted with carbon dioxide, aragonite and calcite produced on the surface in the first 27 min, aragonite slowly transformed into calcite |
| Black et al. [167] | NMR; XRD | C-S-H amorphous calcium carbonate hydrate was formed within minutes of exposure to air |
| Sevelsted et al. [164] | $^{27}$Al NMR | All Al sites consumed during the carbonization reaction, mainly incorporated into the amorphous silicon phase, few Al [5] sites were found |
| Morandeau et al. [172] | Phenolphthalein spray test | High-magnesium cement paste prevented removal of extra calcium from the C-A-S-H gel, preventing the progress of the carbonization reaction |
| Ibáñez et al. [174] | Raman microscope | Carbonization in $C_3S$ 500–1000 μm from surface |

### 6.2. Micro-Nano Structure of C-S-H under Chloride Attack

The chloride attack is one of the main factors affecting the durability of reinforced concrete [176–178]. For instance, chloride ions widely exist in seawater, deicers or directly in binders and may cause serious corrosion of steel bars [179], leading to cracking [180], spalling and a decrease in the bearing capacity of steel bars [181]. The chloride ions penetrate the steel protective layer through the connected pores of cement-based materials. Once the ion concentration on the surface of the steel bar exceeds a certain value, it will destroy the passive film on the steel bar and the corrosion begins, so that the possibility of structural damage is greatly increased. Therefore, the exploration of the mechanism of chloride ion and C-S-H will help improve the performance of reinforced concrete served in the chloride environment.

The binding capacity of the C-S-H was found to depend on its Ca/Si ratio; C-S-H with a high Ca/Si ratio had a great binding capacity [182]. Plusquellec and Nonat aimed to clarify the interaction between C-S-H particles and chloride anion; the main finding was that chlorides did not adsorb on C-S-H particles, although they tended to accumulate in the diffuse layer where they competed with $OH^-$. The device used for their study is presented in Figure 8. The addition of calcium salt increased very slightly the calcium to silicon ratio of the initial C-S-H particles [183]. The results showed that, under the long-term dry–wet cycles and chloride ion coupling erosion, fly ash and slag provided additional $Al_2O_3$ to react with $Cl^-$ and tended to produce Friedel's salt with small interlayer spacing, thereby realizing concrete possessing a high chemical binding capacity with $Cl^-$. The improvement in the physical adsorption capacity of concrete with $Cl^-$, which was due to the formation of additional hydration products of C-S-H gel [184]. XRD/Rietveld refinement method and TGA were performed to unravel the phase assemblages change upon exposure. It was revealed that besides the formation of Friedel's salt, the addition of Na could allow

the enhanced physical binding of chloride as a result of the formation of C-A-S-H, i.e., the substitution of Si by Al in C-S-H gel [185].

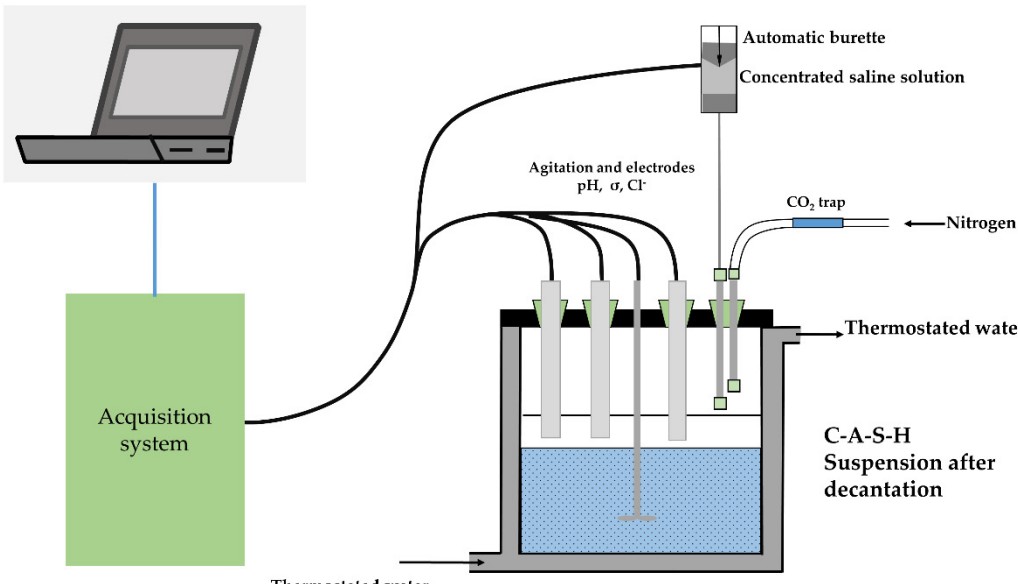

**Figure 8.** Schematic representation of the device developed for in situ analysis of ion adsorption [183].

The chloride bound in C-S-H can be roughly divided into two parts: physical or chemical bonding. Some authors stated this interaction to be physically dominated, while others considered it was a chemical interaction.

Elakneswaran et al. considered the adsorption of chlorine to non-ionized silanol groups [186] and investigated the chloride binding behaviors of cement such as calcium ions. Measured zeta potential of C-S-H as a function of calcium concentration is shown in Table 9 and measured zeta potential of C-S-H as a function of chloride concentration in Table 10. The results show that the monosulfate hydrate and C-S-H phases have significant chloride binding capacities. In addition, the chloride-binding isotherm of C-S-H shows a good fit to a Langmuir-type adsorption. The binding capacity of C-S-H is saturated at about 0.6 mmol/g at a high chloride concentration over 2 mol/L. Based on the chloride binding isotherm of monosulfate hydrate and C-S-H, the chloride binding isotherms of cement pastes are shown to be realistically predicted. [187]. The main factors leading to the change in physical binding capacity may be the Ca/Si ratio of the C-S-H phase and the cations adsorbed to C-S-H or replaced by $Ca^{2+}$ in C-S-H [188]. Exceptionally, foreign $Mg^{2+}$ will replace $Ca^{2+}$ in C-S-H, resulting in some structural changes and a reduction in physical binding capacity [182].

**Table 9.** Measured zeta potential of C-S-H as a function of calcium concentration [186].

| $Ca^{2+}$ Concentration (mmol/L) | $\zeta$ Potential (mV) |
|:---:|:---:|
| 0 | −11.7 |
| 0.1 | −8.5 |
| 0.5 | −6.8 |
| 0.6 | −3.2 |
| 0.7 | −1.2 |
| 0.8 | 2.5 |
| 0.9 | 4.6 |
| 1.0 | 5.1 |
| 2.5 | 15.6 |
| 10.0 | 27.3 |

**Table 10.** Measured zeta potential C-S-H in CH with varying NaCl concentration [186].

| Cl⁻ Concentration (mmol/L) | ζ Potential (mV) |
|---|---|
| 1.0 | −2.8 |
| 2.1 | −3.3 |
| 3.0 | −3.6 |
| 5.0 | −4.4 |
| 10.0 | −5.8 |
| 15.0 | −6.4 |
| 17.0 | −6.9 |
| 20.0 | −7.5 |
| 25.0 | −8.1 |
| 30.0 | −8.5 |
| 35.0 | −9.6 |
| 40.0 | −10.0 |

Regarding chemical interactions, Ramachandran distinguished three different types of interactions between C-S-H particles and chlorides, namely, the chemical adsorption layer on the surface, in the middle layer and tightly bound in the structure [189]. Beaudouin et al. clarified this study by dividing the adsorbed chlorides into two categories: the first category was soluble in water and insoluble in alcohol, which corresponded to anions adsorbed on the surface and intermediate layer. The second one was soluble in alcohol but insoluble in water [190].

Hirao et al. found that chloride-binding isotherm of C-S-H was fitted to the Langmuir-type successfully, and the saturated amount of bound chloride was 0.06 mmol/g at high chloride concentration over 2 mol/L [187]. Yogarajah et al. investigated the absorbed chloride content of synthetic C-S-H and found that it could be adequately described by the Freundlich equation, and the absorbed chloride increased with the chloride concentration increasing. Measured relative surface charge densities of C-S-H as a function of pH are shown in Table 11 [191].

**Table 11.** Measured relative surface charge densities of C-S-H as a function of pH [191].

| pH | Surface Charge Densities (mC/m$^2$) |
|---|---|
| 11.10 | 0 |
| 11.20 | 2 |
| 11.24 | 23 |
| 11.30 | 29 |
| 11.41 | 62 |
| 11.53 | 83 |
| 11.60 | 110 |
| 11.71 | 132 |

Zhou et al. divided the Ca$^{2+}$ in C-S-H into structural surface calcium and interlaminar calcium and carried out a molecular dynamics simulation on the physical adsorption properties of the two kinds of calcium ions to chloride ions. They pointed out that the structural surface calcium had a stronger adsorption capacity for chloride ions than interlaminar calcium. The Ca-Cl cluster formed by interlayer calcium ions after the adsorption of chloride ions only stayed for a short time, and they believed that C-S-H with a high Ca/Si ratio would have lots of non-bridging oxygen and, thus, enable the absorption of massive chloride ions [192,193]. As for the interaction between C-A-S-H and chloride ion, the recent investigation of thermodynamic simulation, TGA and molecular dynamics simulation has shown that the interaction is mainly the chemically weak adsorption on the surface. This weak adsorption is caused by limited defective positive ions on the surface of C-S-A-H and unstable Ca-Cl clusters [194,195].

### 6.3. Micro-Nano Structure of C-S-H under Sulfate Attack

Sulfate attack can also cause damage to cement-based materials, but the damage mechanism is not the same as that of chloride attack. It is equally important to explore the mechanism between sulfate attack and C-S-H.

The damage process of plain and blended cement mortars subjected to sulfate attack under electrical field was investigated. The results showed that electrical field resulted in the dissolution of CH and the decomposition of C-S-H gel due to an accelerated leaching process. The electrical device is demonstrated as Figure 9 [196]. Microstructural experimental results demonstrated that in the three different stages of sulfate attack, degradation of pastes was primarily associated with the migration behavior and bonding configuration of aluminum; in the early ages, Al was mostly present in C-A-S-H, and thus, the damage of pastes hardly appeared, while at later ones, Al had been largely transferred from C-A-S-H into AFt, leading to expansive damage [197]. Taylor et al. utilized Portland cement pastes that had been stored for 6 months in solutions of sodium or magnesium sulfate examined by SEM. The results showed replacement of monosulfate by AFt, which was closely mixed with the C-S-H gel; disappearance of CH, partial decalcification of C-S-H and precipitation of gypsum and AFt [198].

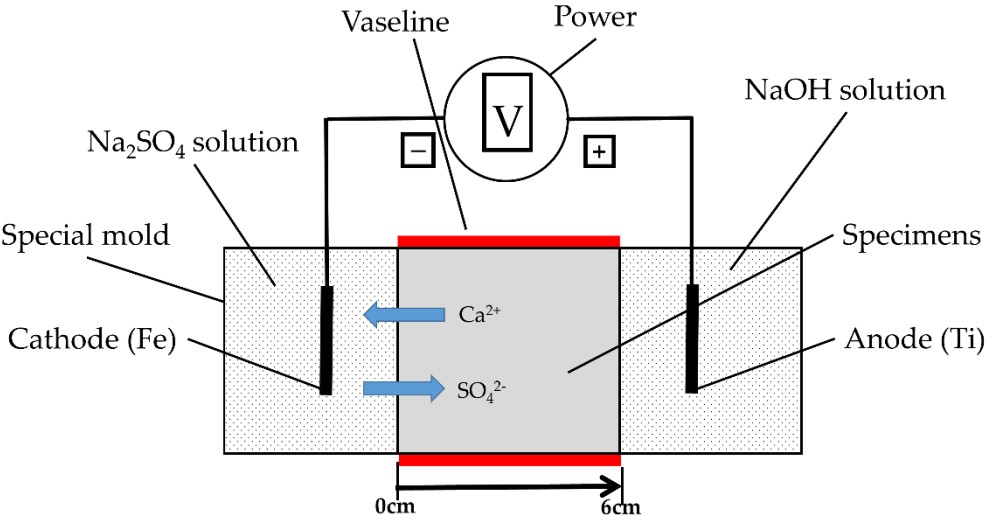

**Figure 9.** Schematic drawing of combined action test of sulfate attack and lelctrical field [196].

The presence of $Mg^{2+}$ made C-S-H gel decalcify, dissolve and decompose, increasing the sulfate attack on the concrete. The deterioration of concrete due to the $MgSO_4$ attack was attributed to decalcification of C-S-H gel to form M-S-H. From the investigation of hydrated cement paste exposed to excessive volume of sea water, M-S-H phase was observed with the Mg/Si ratio $\approx$ 1 and the Al/Mg ratio $\approx$ 0.2, which appeared to have formed by ions exchange of Ca by Mg in the decalcified C-S-H [199–203]. Ding et al. studied the influence of sulfate attack on the micro-structure of C-S-H gel in Portland cement paste and fly ash blended cement paste. The results showed that, in $MgSO_4$ solution, $Mg^{2+}$ promoted the decalcification of C-S-H gel by $SO_4^{2-}$ [204–206].

Kunther et al. studied the influence of the Ca/Si ratio of C-S-H gel phase on the interaction with sulfate ions. It was illustrated that the leaching of Ca from C-S-H gel and CH affected the composition of the pore solution and contributed to the developing crystallization pressure of AFt [207]. Feng et al. immersed C-S-H gel in $Na_2SO_4$ solution and $MgSO_4$ solution, respectively. It was shown that C-S-H gel was attacked by both $Na_2SO_4$ and $MgSO_4$, and gypsum formed whether in a low- or high- concentration solution of $Na_2SO_4$ and $MgSO_4$ [208].

Yang et al. utilized a Clay-FF force field to study the formation mechanism of Ca-$SO_4$ clusters on the surface of two kinds of C-S-H gels and pointed out that the formation of such clusters was dominated by electrostatic forces between the oxygen ions of sulfate and

the defective calcium ions on the surface of C-S-H [209]. Hu et al. studied the interaction between C-S-H and sulfate ions by using NMR and thermodynamic simulation, and the results also showed that the sulfate attack could lead to decalcification and dealumination of C-S-H. Reducing the ratio of Ca/Si or increasing the ratio of Al/Si could effectively improve the thermodynamic stability of C-S-H under the action of sulfate attack [210]. Through the thermogravimetric results, Hu et al. inferred that C-S-H would be decomposed and converted into AFt under the attack of sulfate ions in 200 days. The decomposition process can be roughly divided into three stages: slow, accelerated and stable stages [197]. Table 7 shows the microstructure evolution of C-S-H gels under sulfate attack.

Sulfate attack usually involves the formation of expansive sulphate phases, such as gypsum and AFt, which can cause cracking of mortar and concrete [211–215]. Compared with conventional sulfate attack, cement-based materials containing carbonate sources can undergo thaumasite sulfate attack (TSA) at low temperature, which is considered to be other destructive behavior.

There were two main formation routes of thaumasite below 15 °C [216,217]. One was a direct route from C-S-H reacting with sulfate, appropriate carbonate, $Ca^{2+}$ ions and excess water. The other was woodfordite route from C-S-H reacting with AFt, $CO_3^{2-}$ and excess water. In the hardened cementitious paste rich in AFt, the C-S-H was prone to react with AFt to form thaumasite. Conversely, in areas of the hardened cement paste where there was a scarcity of AFt, C-S-H was likely to react with calcite and/or $CO_2$ and sulfate to form thaumasite [218]. However, a study of Kohler et al. mentioned that thaumasite was not formed by means of the woodfordite route, and direct route was mightily slow or unlikely [219]. In their views, the formation of thaumasite took place on the surface of AFt when the decomposition of C-S-H occurred in cement paste [220]. In addition, Rahman and Bassuoni divided the formation route of thaumasite into the direct route and indirect route, as shown Figure 10. Field and laboratory evidences indicated that the indirect route (triggered by formation of AFt) was quicker than direct route, but the direct route did not diminish. They proposed that the thaumasite formation could continue after the depletion of AFt due to the dissolution of C-S-H [221].

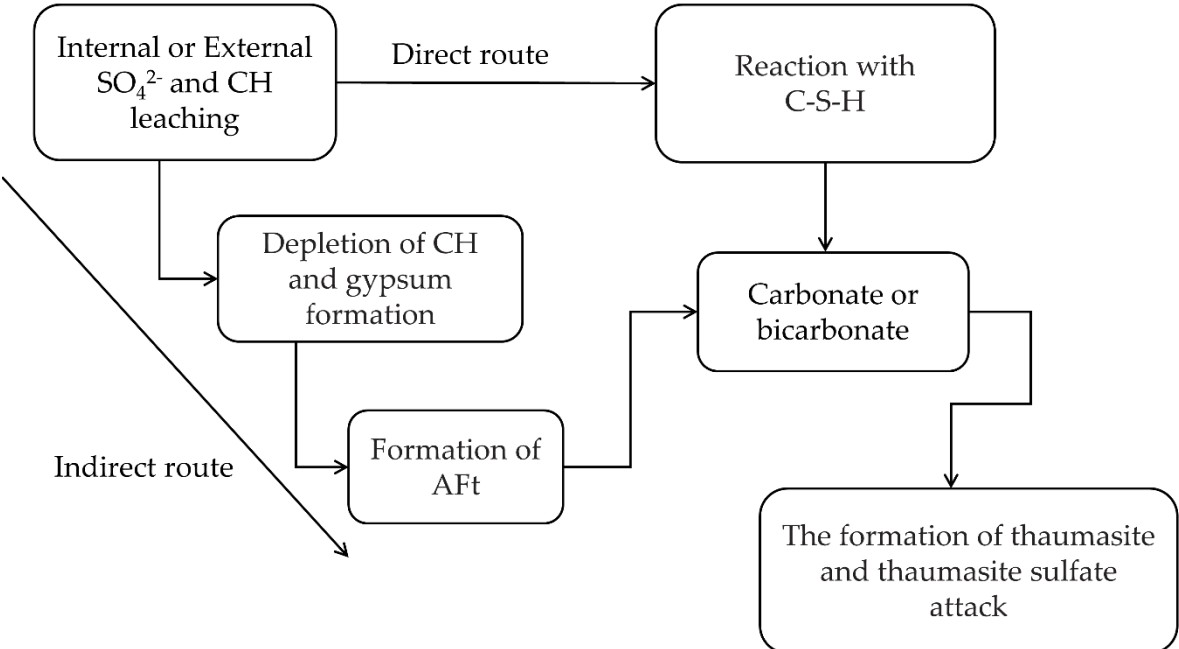

**Figure 10.** A simplified diagram for the direct and indirect routes of the formation thaumasite and thaumasite sulfate attack [221].

Results from thermodynamic calculations indicated that thaumasite can be formed from C-S-H phases with the Ca/Si ratio of 1.7 at extremely low sulphate concentrations. If the Ca/Si ratio in C-S-H phases was reduced to approximately 1.1 via the addition of pozzolanic or latently hydraulic admixtures, these phases could resist high sulphate concentrations, then the formation of thaumasite could be prevented [222]. Figure 11 shows the minimum sulphate ion concentrations required for the transformation of C-S-H phases into thaumasite as obtained by thermodynamic calculation. From Figure 11, it is obvious that the minimum sulphate ion concentration that is required for a transformation of C-S-H phases with Ca/Si ratio of 1.7 into thaumasite is generally low. In contrast, silicon-rich C-S-H phases with Ca/Si ratio of 1.1 show excellent resistance against the formation of thaumasite. Meanwhile, Barbara et al. utilized thermodynamic calculations to indicate that C-S-H phases were stable, while the AFt would most probably be transformed into thaumasite in the long-term and that a pH of about 11 would be reached in the pore solution [223].

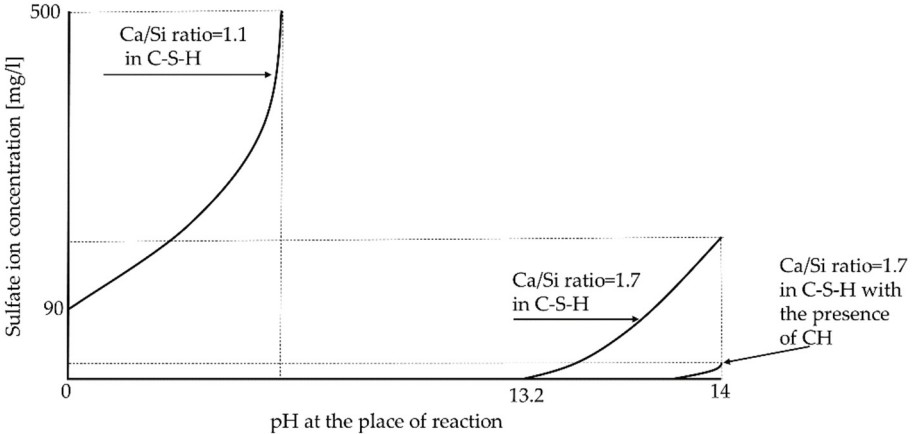

**Figure 11.** Minimum sulphate ion concentrations required for the transformation of C-S-H phases into thaumasite as obtained by thermodynamic calculation [222].

In the presence of CH, the Ca/Si ratio of C-S-H was in the range of 1.6 to 1.8, and calcium-rich C-S-H could lightly be transformed into thaumasite. However, in the absence of CH, the Ca/Si ratio of C-S-H could be reckoned to be between 0.8 and 1.2. Silicon-rich C-S-H showed a high resistance against the formation of thaumasite at moderate sulphate ion concentrations [224].

If few C-S-H and CH were formed in the cement paste, a direct reduction in the volume of thaumasite would be formed. At moderate concentration of sulfate, silicon-rich C-S-H that was formed in the pozzolanic materials showed high resistance to the formation of thauamsite [222,224,225].

## 7. Conclusions

This review has included the body of literature that relates recent research activities to C-S-H. The different preparation methods of C-S-H gel are compared, and the advantages and disadvantages of several preparation methods are analyzed. Meanwhile, in the perspective of different models and molecular dynamics simulations, the structure of C-S-H gels and the structure of C-S-H globules are discussed. In addition, the fractal characteristics and fractal dimension of C-S-H gel are summarized. Additionally, some mechanical properties of C-S-H gel are reviewed. Finally, the durability of C-S-H gels is reported, such as carbonization and chloride/sulfate attacks. Effective and reasonable methods should be explored to reveal the relationship between the structure and mechanical properties of C-S-H.

**Author Contributions:** S.T., Y.W. and Z.G. mostly contributed to the design of the manuscript. X.X. carried out data collection and processing. J.C. and H.A. were involved in the statistical analysis. W.Y. revised the paper. All authors have read and agreed to the published version of the manuscript.

**Funding:** Authors are thankful to the Opening Funds of State Key Laboratory of Building Safety and Built Environment and National Engineering Research Center of Building Technology under grant of BSBE2020–1, National Key R&D Program of China under grant of 2017YFB0310005, National Natural Science Foundation of China under grant of 51602229, Jiangsu Province Natural Science Foundation under grant of BK20181187, Opening Project of State Key Laboratory of Green Building Materials under grant of 2019GBM05.

**Data Availability Statement:** The data that support the findings of this study are available from the corresponding author upon reasonable request.

**Acknowledgments:** The authors would like to thank all the three anonymous referees for their constructive comments and suggestions.

**Conflicts of Interest:** The authors declare no conflict of interest.

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
