# Peer review of "Structure, Fractality, Mechanics and Durability of Calcium Silicate Hydrates"

_fractalfract, doi:10.3390/fractalfract5020047_

Round 1
Reviewer 1 Report
Very well-written review, comprehensive, extensive, and well-rounded - for a review on cement-based materials. However, considering the focus of the journal Fractal and Fractional, the paper is unfortunately lacking in its coverage of the corresponding topics. Essentially, out of the 24 pages of the manuscript (References excluded), only 2 are dedicated to the topic in question (8.3%), with the References directly tied to the topic of fractality being 19 out of 202 (9.4%). I am afraid that this does not justify the publication of this manuscript in this journal - and does not do justice to your manuscript and hard work to publish in such an off-field journal. My proposal is to seek a journal more suitable to material science and cement-based materials/engineering, where a small section on fractality might be more welcomed as an extra topic of interest.
Otherwise, if you insist in publishing here, you will need to invest considerable effort to amend your document by expanding on Section 4 - Fractality. Recommendations include: further input on the mathematical treatment of data obtained by the mentioned methods in order to extract fractal metrics (e.g. box-counting dimension and how it is implemented), physical implications of non-integer dimensions in (Df<>D) on porosity, elucidation of surface fractal dimension (Ds) vs. mass fractal dimension (Dm), surface roughness metrics and tribological considerations etc.
As a side-note: the fractal dimension cannot be evaluated directly; it is a mathematical concept with tentative connection to the physical.
This also is true for the simplest methods based on roughness measurements, i.e. AFM surface analysis. Any such contour/relief data need to undergo a statistical process to extract a fractal dimension (e.g. box counting method) - and this does NOT mean that the structure is self-similar (fractal), only that it can be described by a fractal measure.
Finally, there are minor grammatical-syntactical mistakes throughout the text. I have taken the liberty of correcting these on the attached pdf document, also including other comments in the abstract, introduction, and Section 4 of the manuscript.

Author Response
Very well-written review, comprehensive, extensive, and well-rounded - for a review on cement-based materials. However, considering the focus of the journal Fractal and Fractional, the paper is unfortunately lacking in its coverage of the corresponding topics. Essentially, out of the 24 pages of the manuscript (References excluded), only 2 are dedicated to the topic in question (8.3%), with the References directly tied to the topic of fractality being 19 out of 202 (9.4%). I am afraid that this does not justify the publication of this manuscript in this journal - and does not do justice to your manuscript and hard work to publish in such an off-field journal. My proposal is to seek a journal more suitable to material science and cement-based materials/engineering, where a small section on fractality might be more welcomed as an extra topic of interest.
Response: Your comment is greatly appreciated. We have realized that the corresponding topics of Fractal and Fractional are not comprehensively enough.
Otherwise, if you insist in publishing here, you will need to invest considerable effort to amend your document by expanding on Section 4 - Fractality. Recommendations include: further input on the mathematical treatment of data obtained by the mentioned methods in order to extract fractal metrics (e.g. box-counting dimension and how it is implemented), physical implications of non-integer dimensions in (Df<>D) on porosity, elucidation of surface fractal dimension (Ds) vs. mass fractal dimension (Dm), surface roughness metrics and tribological considerations etc.
Response: The content of “4.Fractality of C-S-H” was extended according to comments. The content includes the principle and application of the box counting method, the influence of surface roughness on cement-based materials and C-S-H gels, and the concept of non-integer dimensions. The manifestation of tribological properties was also included.
As a side-note: the fractal dimension cannot be evaluated directly; it is a mathematical concept with tentative connection to the physical.
This also is true for the simplest methods based on roughness measurements, i.e. AFM surface analysis. Any such contour/relief data need to undergo a statistical process to extract a fractal dimension (e.g. box counting method) - and this does NOT mean that the structure is self-similar (fractal), only that it can be described by a fractal measure.
Response: Your comments are very constructive. We have corrected our wrong understanding of fractal dimensions.
Finally, there are minor grammatical-syntactical mistakes throughout the text. I have taken the liberty of correcting these on the attached pdf document, also including other comments in the abstract, introduction, and Section 4 of the manuscript.
Response: Your comment is significant. All the grammatical-syntactical mistakes were corrected according to comments.
Please see the attachment for the details of revision.

Reviewer 2 Report
The manuscript of the article is of great scientific interest. The research results allow the selection of components in order to control the properties of cement materials.
Author Response
The manuscript of the article is of great scientific interest. The research results allow the selection of components in order to control the properties of cement materials.
Response: We appreciate the reviewer’s positive comments.
Reviewer 3 Report
All material on calcium silicate hydrates is reported perfectly.
Only few editorial remarks can be stated:
- The repetition in the lines 308 - 311.
- The sentence in lines 312 - 313 is not correct.
- The atoms must be specified (designated) in the Fig. 1.
Author Response
All material on calcium silicate hydrates is reported perfectly.
Only few editorial remarks can be stated:
- The repetition in the lines 308 - 311.
- The sentence in lines 312 - 313 is not correct.
- The atoms must be specified (designated) in the Fig. 1.
Response: Reviver’s comments are greatly appreciated. We have examined the manuscript again and again and made some corrections as follows:
- In Lines 308-311, we have deleted the repetition sentence “The density of C-S-H in the hardened cement paste varies with the degree of hydration (α) and water cement ratio”.
- In Lines 312-313, we have changed the improper statement “The density of C-S-H without gel pore water decreased from 2.73 g/cm3 when α≈0.4 to 2.65g/cm3 when α≈0.9; the density of C-S-H including gel water decreased from about 1.8 g/cm3 when α≈0.4 increases to 2.1 g/cm3 when α≈0.9” as “When α increased from 0.4 to 0.9, the density of C-S-H without gel pore water decreased from 2.73 to 2.65g/cm3, the one including gel water increased from about 1.8 to 2.1 g/cm3”.
- In Fig. 1, the atoms have been specified as Ca, Si, O and H.
- Please see the attachment.

Reviewer 4 Report
Based on the high quality of the paper I recommend its publication without changes. I thank authors for really good job they have done within the evaluation of experiments and writing the paper.
Author Response
Based on the high quality of the paper I recommend its publication without changes. I thank authors for really good job they have done within the evaluation of experiments and writing the paper.
Response: We appreciate the reviewer’s positive comments.
Round 2
Reviewer 1 Report
With the provided amendments, the manuscript is better suited for publication in this journal.
There are still some minor errors concerning grammar and syntax - you could give it a final polish before publication.
Author Response
Your comments are very constructive. We have corrected the grammatical errors.